# Disentangling Atmospheric, Hydrological, and Coupling Uncertainties in Compound Flood Modeling within a Coupled Earth System Model

Dongyu Feng[1], Zeli Tan[1], Darren Engwirda[2], Jonathan D. Wolfe[2], Donghui Xu[1], Chang Liao[1], Gautam Bisht[1], James J. Benedict[2], Tian Zhou[1], Mithun Deb[3], Hong-Yi Li[4], and L. Ruby Leung[1]

[1]Atmospheric, Climate, & Earth Sciences Division, Pacific Northwest National Laboratory, Richland, WA, 99354, USA
[2]T-3 Fluid Dynamics and Solid Mechanics Group, Los Alamos National Laboratory, Los Alamos, NM, 87545, USA
[3]Marine and Coastal Research Laboratory, Pacific Northwest National Laboratory, Sequim, WA, 98382, USA
[4]Department of Civil and Environmental Engineering, University of Houston, TX, 77204, USA

*Correspondence to*: Dongyu Feng (dongyu.feng@pnnl.gov)

**Abstract.** Compound riverine and coastal flooding is usually driven by complex interactions among meteorological, hydrological, and ocean extremes. However, existing efforts of modeling this phenomenon often do not integrate hydrological processes across atmosphere-land-river-ocean systems, leading to substantial uncertainties that have not been fully examined. To bridge this gap, we leverage the new capabilities of the Energy Exascale Earth System Model (E3SM) that enable a multi-component framework that integrates coastal-refined atmospheric, terrestrial, and oceanic components. We evaluate compound uncertainties arising from two-way land-river-ocean coupling in E3SM, and track the cascading meteorological and hydrological uncertainties through ensemble simulations over the Delaware River basin and estuary during Hurricane Irene (2011). Our findings highlight the importance of two-way river-ocean coupling to compound flood modeling and demonstrate E3SM's capability in capturing compound flood extent near the coast, with a hit rate over 0.75. Our study shows the growing uncertainties that transition from atmospheric forcings to flood distribution and severity. Furthermore, an Artificial Neural Network based analysis is used to assess the roles of hydrological drivers, such as infiltration and soil moisture, in the generation of compound flooding. The response of compound floods to tropical cyclones (TCs) is found to be susceptible to these often overlooked drivers. For instance, the flooded area could increase by more than twice (~2.4) if Hurricane Irene was preceded by an extreme antecedent soil moisture condition (AMC). The results not only support the use of a multi-component framework for interactive flooding processes, but also underscore the necessity of broader definitions of compound flooding that encompasses the simultaneous occurrence of intense precipitation, storm surge, and high AMC during TCs.

Keywords: Earth System Model, hydrologic modeling, compound flooding, antecedent soil moisture condition

## 1 Introduction

Compound flooding (CF) is a significant and complex hazard encompassing multiple concurrent drivers such as heavy rainfall, storm surges, and rain-on-snow events (Li et al., 2019) that cause severe socioeconomic and environmental damages (Zscheischler et al., 2018). In coastal regions, CF often arises from a complex interplay of meteorological, hydrological, fluvial, and oceanic processes triggered by tropical cyclones (TCs) (Leonard et al., 2014; Bilskie and Hagen, 2018; Hendry et al., 2019; Loveland et al., 2021). Characterized by high wind speeds and low surface atmospheric pressure, TCs can bring intense rainfall over land and significant storm surge above normal tide levels (Fig. 1). CF poses elevated risks compared to single-source pluvial, fluvial, and coastal flooding due to its broader spatial coverage and extended durations (Wahl et al., 2015; Moftakhari et al., 2017). Sarhadi et al. (2024) suggested that the frequency and intensity of CF events would increase by up to fivefold by the end of this century, driven by factors such as intensified TCs and rising sea levels (Feng et al., 2022). This bleak projection highlights the critical need for advanced integrated modeling strategies, aiming to effectively mitigate future flood risks and improve the resilient infrastructure and adaptive community response plans (Bates et al., 2021).

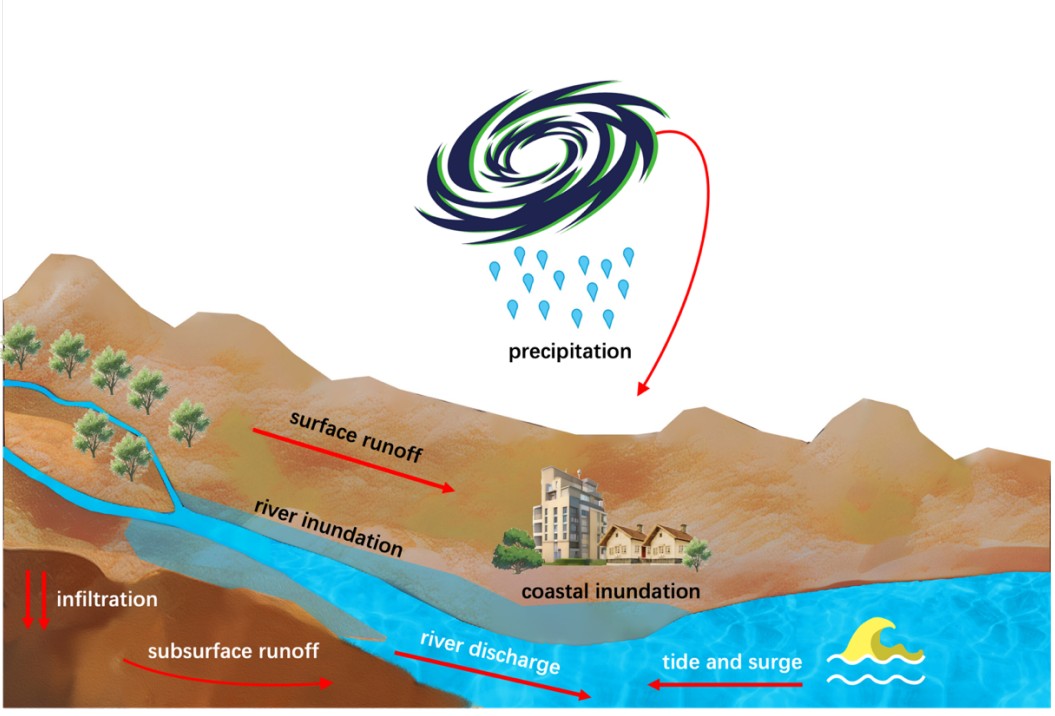

**Figure 1: Compound flooding processes in a coastal river basin during a TC event. This conceptual diagram shows the key elements contributing to CF, resulting from combined riverine and coastal inundation along the river channels and adjacent coastal areas.**

Modeling CF is inherently challenging because it is triggered and impacted by the interactions of processes within multiple Earth system components, including atmosphere, land, river, and ocean, as well as the associated uncertainties (Xu et al.,

2023). Traditional CF modelling is typically based on coupled hydrological models (Feng et al., 2022; Ikeuchi et al., 2017), hydraulic models (Bakhtyar et al., 2020; Bermúdez et al, 2021; Gori et al., 2020b) and hydrodynamic coastal/ocean models (Bennet et al., 2023; Kerns & Chen, 2023; Xiao et al., 2021; Ye et al, 2020) at local, regional and global scales. Over time, more sophisticated methodologies have been developed to enhance CF modeling. These include combined statistical-numerical modeling approaches (Olbert et al., 2023), deep learning (Feng et al., 2023a; Muñoz et al., 2021), data assimilation (Muñoz et al., 2022), reduced-physics ocean models (Eilander et al., 2023; Leijnse et al., 2021), new compound inundation models (Santiago-Collazo et al., 2024), and two-way river-ocean model coupling (Bao et al., 2022; Bao et al., 2024; Feng et al., 2024; Shen et al., 2024; Zhang et al., 2024). The CF modeling uncertainties can be sourced from model structures, parameters, input data, boundary and initial conditions (Abbaszadeh et al., 2022; Beven et al., 2018; Fan et al., 2021). These uncertainties may also cascade through the system (Meresa et al., 2021; Hasan Tanim & Goharian, 2021) and their contributions change dynamically over time (Muñoz et al., 2024).

A recently developed approach is the use of fully coupled Earth System Models (ESMs) to simulate compound flooding (Feng et al., 2024; Zhang & Yu, 2024). By integrating multiple earth system components in a single, tightly coupled framework, ESMs allow for predictive understanding of multi-scale flow processes and their interactions with other relevant processes involving heat, energy, biogeochemical and sediment transport, as well as their impacts on Earth's climate (Ward et al., 2020). Feng et al. (2024) performed the first fully-coupled ESM simulation for CF using the Energy Exascale Earth System Model (E3SM), by integrating recent advancements in E3SM including regionally refined unstructured meshes for atmosphere, land/river and ocean components in the global domain (Deb et al., 2024; Feng et al., 2022), two-way online land-river-ocean coupling (Xu et al., 2022b; Feng et al., 2024), and a 2-dimensional (2D) barotropic ocean model (Lilly et al., 2023).

While state-of-the-art ESMs are being implemented to simulate local extremes, this advancement can inevitably introduce uncertainties. Compared with regional simulations using prescribed atmospheric forcing derived from observation or reanalysis datasets, the model-simulated atmospheric forcings are more uncertain (Hersbach et al., 2020). Atmospheric forcing has critical impacts on the flood simulation (Cloke and Pappenberger, 2009; Hjelmstad et al., 2021, Xu et al., 2025). Specifically, the river discharge intensity, storm surge levels, and CF inundation extents are directly influenced by the TC's track and intensity, as well as the rainfall rate and timing (Gori et al., 2020a; Pappenberger et al., 2005; Zhong et al., 2010). These factors are the primary drivers of the riverine and coastal flooding dynamics. The uncertainty originated from atmospheric forcings would propagate to land, river, and ocean components through the multi-component framework (Deb et al., 2023; Blanton et al., 2020; Joyce et al., 2018). Likewise, the hydrological uncertainties in the land and river components (Giuntoli et al., 2018; Feng et al., 2023b) and the model coupling schemes can also propagate and even amplify the uncertainties. However, the cascading meteorological uncertainty has not been systematically estimated for CF modeling (Abbaszadeh et al., 2022; Xu et al., 2023). It remains unclear whether such uncertainty will amplify or diminish when constrained by the physical processes inherent in ESMs.

The cascading uncertainty in ESMs becomes even more complex with two-way interactive model coupling. In online two-way coupling, a downstream model, while receiving data from its upstream component, sends back real-time computed information at predefined time intervals, enabling bi-directional data exchange. For instance, a river model may send floodplain inundated water extent to the land surface model for estimating flood water infiltration on the floodplain (Xu et al., 2022b). Similarly, an ocean model provides its predicted water levels (Bao et al., 2022; Bao et al., 2024; Feng et al., 2024), velocities (Zhang et al., 2024) or fluxes (Shen et al., 2024) to the river model for capturing the backwater effect. While other uncertainties have been extensively discussed (Camacho et al., 2015; Feng et al., 2019; Muñoz et al., 2024; Willis et al., 2019), the uncertainty relevant to the two-way model coupling has rarely been explored because the coupling capabilities have only recently been developed. Questions are raised regarding the role of model coupling and the magnitude of related uncertainty compared to meteorological uncertainty, especially given the characteristic spatiotemporal scales invoked in land-river and river-ocean coupling. Addressing these questions is critical to refining the performance of interactively coupled ESMs, which is essential for achieving a more comprehensive understanding of the complex interactions and uncertainties associated with CF simulations. Moreover, assessing the enhancements provided by the two-way coupling schemes sheds light on the application of these couplings in future scenarios.

Furthermore, the cascading uncertainty changes with the variability and complexity of hydrological drivers represented in models, because these factors are critical for determining how precipitation is partitioned into runoff and infiltration. As rainfall initially infiltrates the soil, subsurface runoff moves slowly through the soil layers. When the rainfall intensity exceeds the soil's absorption capacity, saturation-excess water leads to surface runoff. The rate of infiltration, which determines the balance between surface and subsurface runoff, is influenced by soil properties, antecedent moisture conditions (AMC) (Ivancic and Shaw, 2015), and land cover types. The runoffs are then routed through river networks, resulting in high river discharge (Fig. 1) (Bevacqua et al., 2020). Understanding the hydrological drivers, including the sensitivity of flood responses to various conditions such as different AMC and rainfall scenarios, is crucial (Tramblay et al., 2010). These factors provide key insights for predicting different flood scenarios (Miguez-Macho and Fan, 2012; Schrapffer et al., 2020). In particular, AMC plays a critical role in the generation of peak runoff and modulating riverine flooding characteristics during heavy precipitation events (Berghuijs et al., 2019; Nanditha and Mishra, 2022). A saturated AMC can significantly amplify flood impacts compared to drier conditions. The relative importance of rainfall and AMC varies depending on the watershed area. Soil moisture becomes a more dominant factor in larger watersheds (Ran et al., 2022). However, the role of these hydrologic drivers in cascading uncertainties sourced from atmospheric forcing has not been thoroughly investigated in the context of CF, partly due to the absence of a tightly coupled modeling system (Jalili Pirani and Najafi, 2020) or insufficient investigation into hydrological processes (Lin et al., 2024). Although Bilskie et al. (2021) and Santiago-Collazo et al. (2024) highlighted the critical consequence if CF is preceded by an antecedent rainfall event, their implementation of the rain-on-grid method does not account for the hydrological processes, such as runoff generation. Addressing these processes would require a detailed hydrological model or land surface component of ESMs. The fully coupled E3SM provides a feasible framework for quantifying the hydrological uncertainties in the CF modelling.

The above-mentioned uncertainties are complicated but must be carefully evaluated for ESMs as they will be more frequently applied for CF simulations in the context of climate change. A variety of approaches have been adopted for understanding the uncertainties of CF modeling. These approaches offer trade-offs between computational cost, physical interpretability, and the ability to disentangle complex drivers. Ensemble-based methods remain a primary strategy for characterizing the cascading uncertainty from the forcing data (Hamill et al., 2011; Hou et al., 2017; Villarini et al., 2019). Multiple realizations with perturbed initial conditions and/or model physics represent a range of scenarios that evolve differently based on the dynamics of the models (Blanton et al., 2020; Saleh et al., 2017; Nederhoff et al., 2024; Wang et al., 2024). Probabilistic frameworks, such as Bayesian inference (Beven and Binley, 1992), provide more robust treatment of parameters and model uncertainties (Naseri & Hummel, 2022), but often rely on strong assumptions and intensive sampling. Machine learning techniques have been increasingly applied to flood modeling (Hu et al., 2019) and are effective at capturing nonlinear relationships of CF drivers (Muñoz et al., 2024), though they require large training datasets and may sacrifice physical interpretability (Shen et al., 2023). Structural equation modeling (SEM; Wright, 1921) has also been adopted to disentangle complex, interacting processes (Du et al., 2015; Santoro et al., 2023). SEM offers a balance between statistical rigor and interpretability in multi-driver systems without a significantly amount of data. Despite these advances, uncertainty quantification within fully coupled ESM frameworks remains relatively underexplored due to high computational demands and limited methodological integration across domains.

This study focuses on exploring and disentangling the atmospheric, hydrological, and coupling uncertainties of coastal CF modeling within the coupled E3SM framework. We first provide a comprehensive description of the physical processes during a TC-induced CF event. We then evaluate the model coupling uncertainties and the cascading meteorological uncertainty using a simulation ensemble of a specific TC event. Using the atmospheric ensemble as a basis, we generated an expanded ensemble and proposed a new machine learning approach to analyze the relative contributions of different hydrological drivers to CF and how these contributions affect the accuracy and reliability of CF simulations over time. Finally, various hydrological and meteorological scenarios are used to delineate a spectrum of plausible CF outcomes in the designated region.

## 2 Materials and Methodology

### 2.1 Model Configuration

This study uses a recently developed configuration in the Energy Exascale Earth System Model (E3SMv2) (Feng et al., 2024). E3SM represents a significant advancement in Earth System modeling (Golaz et al., 2019, 2022). As a fully coupled ESM, E3SM supports dynamic exchanges and propagation of information across its different components. Additionally, several other developments have been recently implemented to further improve the modeling of coastal extremes, including the introduction of high-resolution regional-refined unstructured meshes in global river models (Feng et al., 2022; Liao et al., 2022, 2023a, b), the implementation of interactively coupled land-river-ocean models (Xu et al., 2022b; Feng et al., 2024),

and the global tide model with a wetting and drying scheme in the ocean component (Barton et al., 2022; Pal et al., 2023). Compared with regional models that may provide more detailed inundation at the street level (Costabile et al., 2023; Ivanov et al., 2021), E3SM excels at coupling processes across various earth system components. This capability is crucial for
capturing the complex responses of earth systems to climate change and projecting climate-driven flood hazards.

The new E3SM configuration (hereafter "E3SM coastal configuration") integrates the global three-dimensional (3D) E3SM atmospheric model (EAM), one-dimensional (1D) E3SM land model (ELM), 1D E3SM river model MOSART (MOSART: Model for Scale Adaptive River Transport), and the two-dimensional (2D) barotropic version of E3SM ocean model MPAS-O (MPAS-O: Model for Prediction Across Scales ocean model) (Fig. 2a). This configuration uses three different variable-
155 resolution meshes to improve the E3SM's capability in modeling coastal processes. The EAM mesh features a global resolution of 100 km, with enhanced refinement to about 25 km over the North Atlantic Ocean and eastern North America. Both ELM and MOSART use a land mesh with a coarse resolution of 60 km globally, which is further refined to 30 km across the contiguous US and to 3 km within the Mid-Atlantic watersheds. The MPAS-O mesh offers the highest resolution of 250 m along the US East Coast, specifically designed to capture estuary dynamics, with a broader global resolution of
160 around 1 km everywhere else. The global ELM–MOSART simulations are computationally efficient, requiring less than 10 minutes using 400 CPUs, while the fully coupled ELM–MOSART–MPAS-O simulations take approximately 5 hours. The global bathymetry data of MOSART and MPAS-O are sampled from the 90-m HydroSHEDS digital elevation model (DEM) (Lehner et al., 2008) and the 450-m GEBCO dataset (IOC and IHO, 2020), respectively. The river networks and flow directions are derived using HexWatershed which performs hybrid depression filling and stream burning for river routing in
unstructured meshes (Liao et al., 2022, 2023a, b). The river bankfull width and depth were derived using the power law function with bankfull discharge (Andreadis et al., 2013).

The novel two-way hydrological coupling between land and river components enables E3SM to capture the infiltration of inundated river water in floodplains and, subsequently, the enhancement of subsurface runoff and evapotranspiration from saturated floodplain soils (Xu et al., 2022b). The two-way river-ocean coupling was developed for E3SM to better represent
the dynamic interaction between rivers and oceans, especially during CF events (Feng et al., 2024). This new approach allows for an accurate representation of coastal backwater effects and the mutual influences of river discharge and ocean sea surface height (SSH), providing a more realistic assessment of CF hazards (Feng et al., 2022).

Using the E3SM coastal configuration, we first simulated Hurricane Irene, a TC event that occurred in August 2011 and had large flooding impacts across the Mid-Atlantic region (Fig. 2b). Irene led to significant riverine and coastal flooding in the
175 Delaware River Basin (DRB) and Delaware River Estuary (DBE) due to concurrent intense precipitation and storm surge. Following Feng et al. (2024), an ensemble of 25 EAM simulations with perturbed model parameters were performed to reproduce Irene and associated meteorological outcomes. EAM is initialized from ECMWF Reanalysis v5 (Hersbach et al., 2020) at 00:00Z 26 August 2011. Atmospheric nudging is not applied. The EAM ensemble can reproduce the TC characteristics, including the storm track and intensity (see Appendix A in Deb et al. (2024)). These "prerun" EAM
simulations were then prescribed within E3SM to drive the land, river, and ocean components. EAM (in "data mode")

provides atmospheric forcing to ELM and MPAS-O at a 15-min frequency. MOSART is interactively coupled with ELM and MPAS-O at the 1-hour interval via the E3SM coupler (Craig et al., 2012). The model outputs are archived at 15 minutes for EAM and hourly for ELM, MOSART and MPAS-O. We spun up ELM and MOSART from a 10-year historical simulation forced by Global Soil Wetness Projects version 3 (GSWPv3; Kim, 2017), and MPAS-O from a 1-month simulation with the global tide model. MOSART was validated against the streamflow measurements at 6 USGS gauges along the Delaware River main channel with averaged coefficient of determination ($r^2$) of 0.79 and Kling–Gupta efficiency (KGE; Gupta et al., 2009) of 0.84. MPAS-O was assessed for water level at 6 NOAA tidal gauges across the DBE, showing an averaged $r^2$ of 0.72 and root mean squared error (RMSE) of 0.41 m. Please refer to Feng et al., (2024) for a more detailed description of the E3SM configuration and the model evaluation.

Fluvial and coastal inundations are simulated in MOSART and MPAS-O, respectively. The riverine inundation in MOSART is simulated using a macroscale inundation scheme that assumes the inundation occurs from the lower elevation to higher elevation within each grid cell (Luo et al., 2017; Yamazaki et al., 2011). Coastal inundation simulated on the MPAS-O inland mesh is aggregated onto the coarser MOSART mesh in the DRB. Within each MOSART grid cell, the inundation fraction is determined by the percentage of MPAS-O cells with a simulated water depth over 1 m. This threshold does not imply a spatially uniform adjustment of the GEBCO bathymetry data used by MPAS-O. Instead, it serves as a practical criterion to mitigate biases arising from upscaling inundation extents from the higher-resolution MPAS-O mesh to the coarser MOSART grid. Whenever there is a discrepancy between the inundation area from MOSART and MPAS-O in their overlapped cells near the coastline, the MPAS-O inundation is considered more accurate and will be used.

Here, the total simulated inundation extent of Irene is benchmarked against a 250-m resolution inundation extent dataset based on satellite imagery (Tellman et al., 2021). The dataset is aggregated onto the MOSART mesh for comparison. Within each MOSART cell, we compute the fraction of the observed inundation. The model performance is evaluated using flood metrics defined by Wing et al. (2017), including hit rate ($HR$), false rate ($FR$) and success index ($SI$)

$$HR = \frac{M_1 B_1}{M_1 B_1 + M_0 B_1}, \tag{1}$$

$$FR = \frac{M_1 B_0}{M_1 B_0 + M_1 B_1}, \tag{2}$$

$$SI = \frac{M_1 B_1}{M_1 B_1 + M_0 B_1 + M_1 B_0}, \tag{3}$$

where $M$ and $B$ are the pixels (or grid cells) from model simulations and benchmark data, respectively. The subscripts 1 and 0 represent wet (inundated) and dry cells, respectively. For all the three metrics, a score of 0 indicates poor performance, while a score of 1 represents perfect model performance. In our simulations, a wet cell is identified if the simulated inundation fraction is above a small unitless threshold of 0.02. This threshold minimizes the influence of cells that may only be marginally inundated—likely due to data and model uncertainties—thus ensuring a more reliable assessment of flood extent. The predicted flooded area (FA) is calculated by multiplying the flooded fraction by the corresponding cell area.

## 2.2 Model Coupling Uncertainty

The model coupling uncertainty is evaluated using three experiments (Table 1). The first two experiments implement one-way and two-way coupled ELM and MOSART, respectively, while the third experiment interactively couples MPAS-O with MOSART. All experiments are driven by the same EAM ensemble atmospheric forcing. The MOSART and MPAS-O simulated inundation extent is first evaluated against the benchmark data to justify the necessity of considering both riverine and coastal flooding within the coupled ESM. We then compared the streamflow along the Delaware River mainstem and riverine inundation in DRB in terms of flood metrics among different experiments to demonstrate the uncertainty of two-way coupling. The comparison of riverine inundation between Experiments 1 and 2 and between Experiments 1 and 3 shows the uncertainty from two-way land-river and river-ocean coupling, respectively. The comparison of total inundation between Experiments 1 and 3 quantifies the uncertainty if the ocean component is neglected in the CF simulation.

**Table 1 Numerical experiments for quantifying model coupling uncertainty.**

| Experiment # | Configuration | Flooding type |
|---|---|---|
| 1 | ELM → MOSART | riverine |
| 2 | ELM ↔ MOSART | riverine |
| 3 | ELM → MOSART ↔ MPAS-O | riverine & coastal |

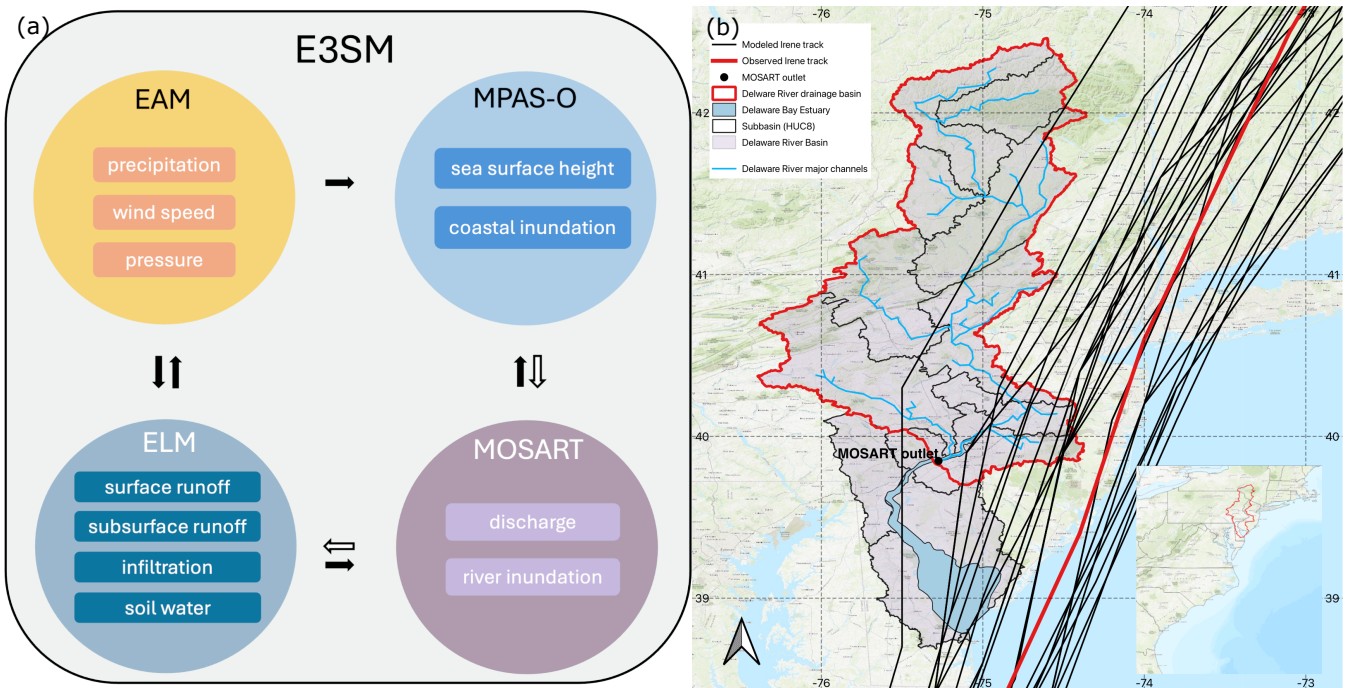

**Figure 2: (a) The multi-component E3SM framework and drivers used for analyses within each model component. The black arrows represent the data flow via the one-way coupled framework. The white arrows are the new flow directions from the 2-way**

land-river and river-ocean models. (b) Map of Delaware river basin (DRB), Delaware bay estuary (DBE), and the observed (red) and modeled (black) Irene tracks. The topographic map in (b) is from the ESRI world topographic map (ESRI, 2012).

### 2.3 Cascading Meteorological Uncertainty

230 To understand the evolution of the meteorological uncertainty cascaded from atmospheric simulations through the multi-component framework, we applied the configuration of Experiment 3 (Table 1) and analyzed the interactions of those physically interconnected variables from the atmosphere, land, river, and ocean components of E3SM (Fig. 2a) including precipitation ($precip$), air pressure ($P_{air}$) and wind speed ($U_{wind}$) from EAM; surface runoff ($Q_{sur}$), subsurface runoff ($Q_{sub}$), infiltration ($Q_{infl}$) and soil water storage ($Q_{soil}$) from ELM; river discharge ($Q$) and riverine inundation area ($A_{river}$)

235 from MOSART; SSH and coastal inundation area ($A_{ocean}$) from MPAS-O. The flux and state variables are represented by their event-accumulated and event-peak values within the Delaware River drainage basin, respectively (Fig. 2b). The estimated relationship between these variables represents the impact of one E3SM component on another component. For MOSART and MPAS-O, due to two-way river-ocean coupling, mutual relationships can occur between the related variables. The magnitude of uncertainty amplification or diminishment is quantified using normalized median absolute deviation

240 (NMAD):

$$\text{NMAD} = \frac{\text{median}(|X_i - \text{median}(X)|)}{\text{median}(X)}, \tag{4}$$

and coefficient of variation (CV)

$$\text{CV} = \frac{\sigma}{\mu}, \tag{5}$$

where $X_i$ represents a variable $X$ modeled at the $i$th ensemble run, and $\mu$ and $\sigma$ are the mean and standard deviation of the

245 corresponding variable computed from all ensemble simulations. These two metrics measure the spread of simulations with respect to the ensemble median and mean values separately.

Additionally, structural equation modeling (SEM) is applied as a path analysis method (Wright, 1921) to trace the flow of data and uncertainty from the 25 ensemble members. SEM estimates the complex relationships between two groups of variables by fitting multivariate regressions and uses the coefficient of a predictor to represent its contribution to the

250 response variable. The Python library *semopy* is used in our SEM analyses (Igolkina and Meshcheryakov, 2020).

### 2.4 Uncertainty of Hydrological Drivers

The hydrological drivers we selected for uncertainty analysis include surface runoff, subsurface runoff, infiltration, and soil water storage. As the influence of these hydrological drivers shifts throughout a TC event due to changes in precipitation patterns, we chose to examine the cumulative impacts of these drivers across the entire event and track the temporal

255 evolution of each driver's influence. For this purpose, we expand the original EAM simulation ensemble by introducing variations in AMC and runoff generation parameters within ELM. This expanded ensemble enables us to apply a machine learning approach to compute the permutation importance of each hydrological driver, providing insights into their roles in modulating flood exposure. We focus exclusively on riverine flooding in this analysis. To avoid the substantial

computational burden associated with MPAS-O, we impose MPAS-O simulated water level as the coastal boundary condition of MOSART (Feng et al., 2022). This approach represents the coastal backwater effects during CF, comparable to those obtained from the two-way river-ocean coupled configuration in Experiment 3 (Feng et al., 2024).

### 2.4.1 Expanded Ensemble simulation

The original EAM ensemble is expanded by first selecting 5 ensemble members whose river discharge and precipitation values span the full range of the ensemble and are approximately evenly spaced across that range during Hurricane Irene (Fig. S1 and S2) and then running each member with multiple AMC scenarios and different sets of runoff generation parameters. Five AMC scenarios were chosen to reflect a broad range of hydrological drivers based on historical soil moisture trends, spanning from the driest to the wettest states. Specifically, we used the 0th, 25th, 50th, 75th and 100th percentiles of basin-averaged soil moisture ($0.067 \sim 0.087$ kg m$^{-2}$) during hurricane seasons from 2005 to 2011 as modeled in a historical ELM simulation (Fig. S1). The AMC at 75th percentile aligns with the observed AMC of Irene. Two parameters in ELM ($f_{over}$ and $f_{drain}$), that determine the runoff generation are considered, where $f_{over}$ determines the saturation fraction, i.e. how much surface runoff is generated from precipitation, and $f_{drain}$ controls the subsurface runoff generation (see Appendix A for a detailed definition). Runoff is highly sensitive to both $f_{over}$ and $f_{drain}$, which values usually have to be determined through sensitivity analysis. In the Mid-Atlantic region, as suggested by Xu et al. (2022a), we selected $f_{over}$ values at 0.1, 0.5, 1, 2.5 and 5, and $f_{drain}$ values at 2, 2.25, 2.5, 3 and 5. The varying peak discharge observed in the main channel of the Delaware River and the extent of riverine flooding indicate a broad range of possible scenarios captured in simulations that use the perturbed parameters of atmospheric conditions, AMC, and the parameters $f_{over}$ and $f_{drain}$ (Fig. S2 and S4~S6). Using the selected 5 ensemble members, 5 AMC scenarios, 5 values each for $f_{over}$ and $f_{drain}$, we performed a total of 625 ensemble simulations.

### 2.4.2 Quantifying Hydrological Driver Importance

To quantify the relative importance of each hydrological driver of CF, we employed a two-stage Artificial neural network (ANN) approach (Fig. 3), trained and tested using data from all 625 ensemble simulations. Compared to traditional regression models, ANN is particularly advantageous for capturing the complex, nonlinear relationships that exist between the diverse hydrological drivers and the resulting impacts on river systems (Goodfellow et al., 2016; LeCun et al., 2015; Tsang et al., 2017).

The first ANN model emulates the relationships between the hydrological drivers of $Q_{sur}$, $Q_{sub}$, $Q_{infl}$ and $Q_{soil}$ and perturbation parameters. Here, the input features are precipitation, AMC, $f_{over}$ and $f_{drain}$, and the outputs are the aforementioned hydrological drivers. Then, these outputs become the input features for the second ANN, which emulates the relationships between river discharge and inundation area and these input features. To perform a detailed analysis, we first assessed the event-accumulated impacts of these drivers by aggregating data over the entire TC event. We also examined

fine temporal impacts by using the second ANN on a daily basis. This allows us to understand not only the overall effect of each driver but also their day-to-day variations throughout the event. The relative importance of the input features on the output features is quantified using permutation importance. For more details about the ANN model setup and permutation importance calculation, please refer to Appendix B.

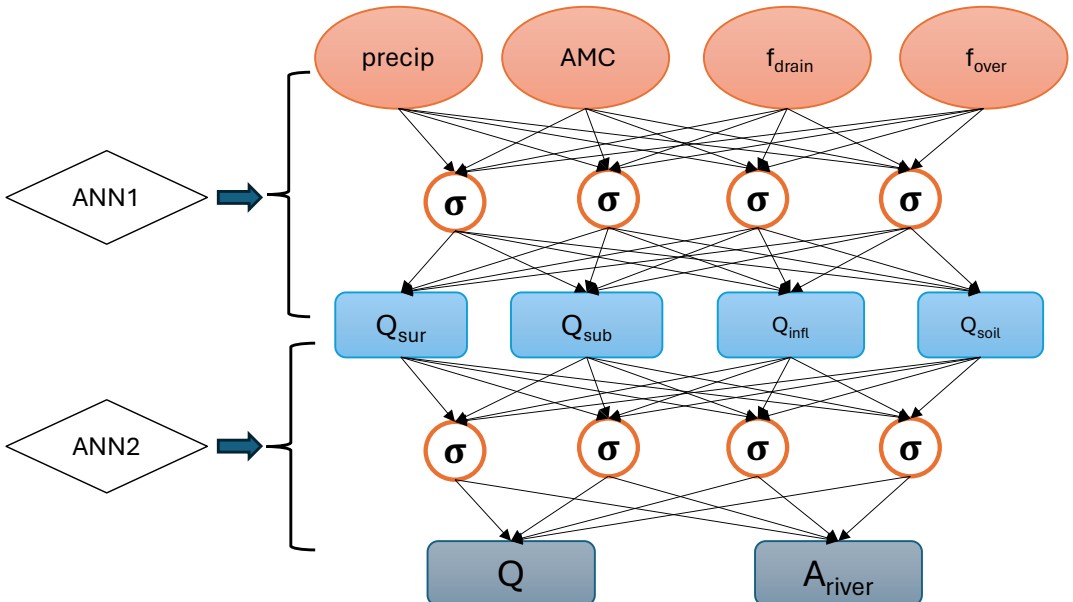

**Figure 3: The densely connected ANNs for quantifying the relative importance of hydrological drivers to river discharge $Q$ and inundation area $A_{river}$. Only 4 neurons per hidden layer are shown for illustration purposes. AMC refers to antecedent soil moisture condition.**

## 3 Results

### 3.1 Model Coupling Uncertainty

In Experiment 3, the E3SM coastal configuration employs the coupled MOSART and MPAS-O models to simulate compound riverine and coastal inundation (Fig. 4). The results indicate that MOSART can predict riverine flooding along the lower Delaware River and its upstream tributaries. However, the model tends to overestimate the maximum extent of flooding along the Delaware River mainstem and some tributaries (Fig. 4a). Occasionally, some observed inundated cells in the upstream are captured by the model. Such bias is likely caused by the coarse spatial resolution of the river mesh,

inaccurate river network delineation, and missing processes such as damming and flood defense constructions. Despite refinement, the mesh and river network still do not achieve the detail provided by regional high-resolution models (Dullo et al., 2021). More importantly, although ELM–MOSART simulates extensive riverine inundation along the Delaware River mainstem and tributaries through precipitation-induced runoff (Fig. 4a), it does not capture inundation in low-lying shoreline areas near the coastline. This is because tide and storm surge that elevated local water levels sufficiently to exceed the

inundation threshold in coastal cells are not included in this configuration. MOSART's macroscale inundation scheme does not simulate lateral water propagation across grid cells, and coastal inundation requires dynamic oceanic forcing. By integrating MPAS-O (Fig. 4b), which includes two-dimensional wetting and drying, the model captures these near-coastline inundations more accurately.

Comparison of flood metrics also confirms the importance of incorporating both riverine and coastal dynamics through a
river-ocean coupled configuration (Fig. 5). Compared to Experiment 1 (Table 1) which does not activate MPAS-O, the river-ocean coupled configuration in Experiment 3 remarkably improves *HR* and *SI* by twofold with more than doubled the predicted flooded area (*FA*) and reduces *FR* by ~0.1. The change in flood metrics implies that a significant portion (>70%) of the compound flooded area during Irene is accounted for by coastal flooding, which could be otherwise neglected if the ocean model is not coupled. However, the integration of MPAS-O does not reduce the MOSART-overpredicted flooded
regions significantly, as suggested by the change of *FR*. The overestimation in *FR* is likely due to the bias in the MODIS satellite data, the macroscale inundation scheme in MOSART, and the MOSART mesh resolution. The flood extent dataset (Tellman et al. 2021) could underestimate the actual flooding area due to the uncertainty in the cloud cover removing technique (Zhang & Yu, 2024). Its fidelity further decreases in the upstream direction due to the existence of vegetation covers (Sexton et al., 2013). In addition, the macroscale inundation scheme may not capture the subgrid connectivity given
the grid resolution of 5 km (Xu et al., 2022b). However, these findings highlight the synergistic nature of river and ocean modeling in improving CF simulations in E3SM.

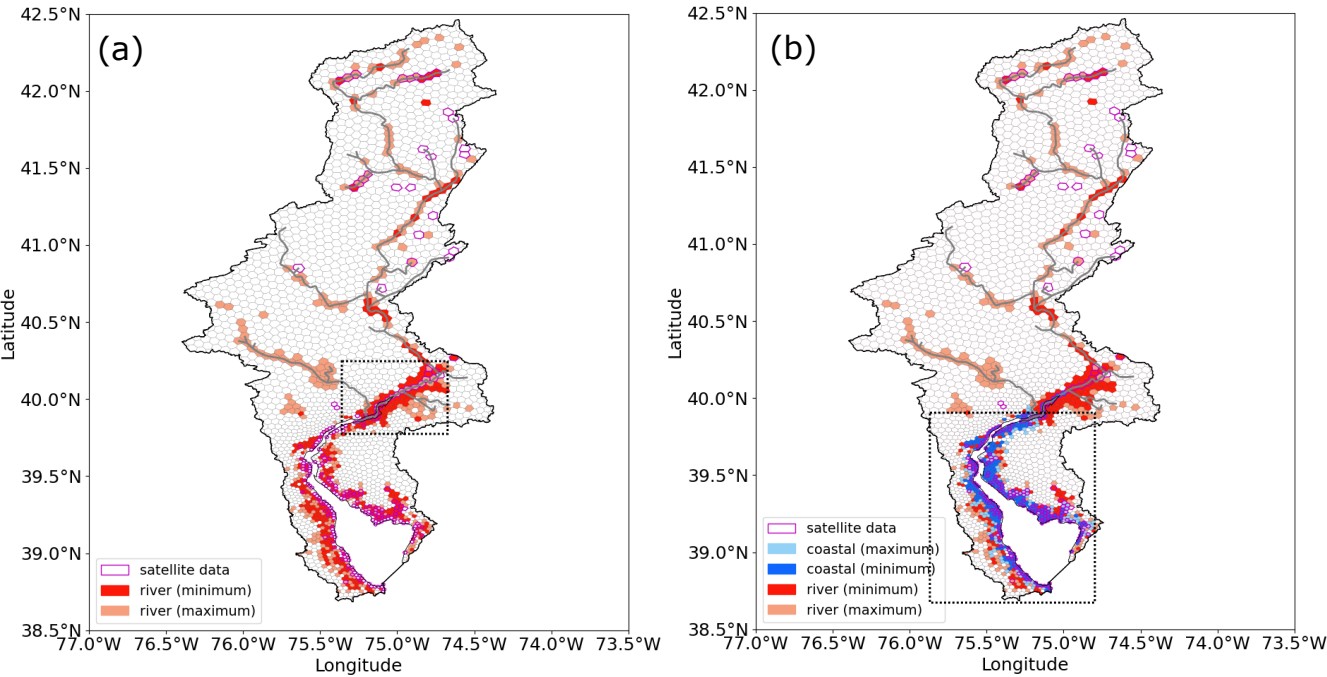

**Figure 4: (a) MOSART simulated riverine inundation (red) against satellite-measured inundation (magenta box). The black dashed box highlights the lower Delaware River reach. (b) E3SM simulated riverine (red) and coastal (blue) total inundation**

against satellite data (magenta box). The black dashed box represents the coastline of DBE where extensive coastal inundation
occurred. In both panels, dark and light colors represent the minimum and maximum inundated extent from the ensemble
simulations, respectively. The gray lines are the major river channels.

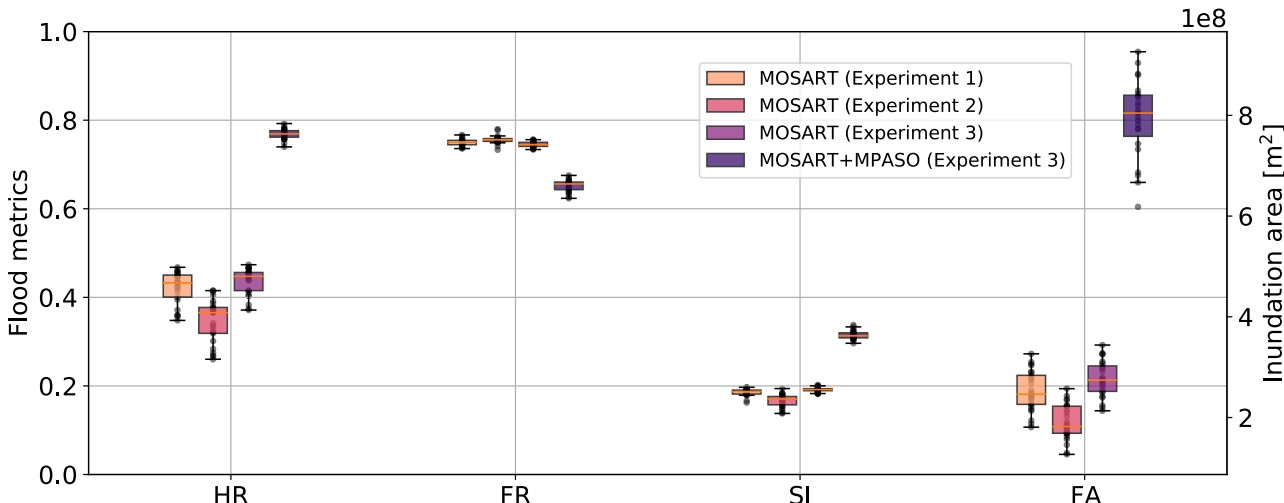

Figure 5: Flood metrics of hit rate (*HR*), false rate (*FR*), success index (*SI*) and flooded area (FA) used to compare riverine
flooding in Experiments 1~3 and the combined riverine and coastal flooding in Experiment 3. Experiment 1 and 2 include land
and river components, while Experiment 3 runs all land, river and ocean components (Table 1). MOSART only considers riverine
inundation, while MOSART+MPASO accounts for both riverine and coastal inundation. Whiskers extend to 1.5 times the
interquartile range from the quartile boundaries.

The comparison of Experiments 1~3 (Table 1) demonstrates the distinct role of land-river-ocean coupling in influencing CF
(Fig. 6). Specifically, the implementation of two-way land-river coupling leads to a noticeable decrease in peak discharge
along the Delaware River mainstem by 10~50 m$^3$/s which slightly increases towards the river outlet (Fig. 6a). Consequently,
the simulated flooded area across the watershed is reduced in Experiment 2 compared to Experiment 1 (Fig. 6b). These
reductions, despite being sporadic in upstream regions, are predominantly observed in the Lower Delaware River reach and
near the coastline (Fig. 6b). This expected change is attributed to the two-way interaction of land and river hydrology
implemented in Experiment 2, in which floodplain inundated water from MOSART is transferred to ELM, thereby reducing
water storage within the channel and flood extent (Luo et al., 2017). Conversely, the influence of two-way river-ocean
coupling (Experiment 3) appears to be mainly confined to the river reaches close to the outlet (Fig. 6c), where it significantly
increases local streamflow (Fig. 6a). This is a result of more accurately representing the water and momentum fluxes
between the river and ocean as well as coastal backwater effects. The elevated water levels due to tide and storm surge force
an upstream propagation of ocean water into the river channel, resulting in a local increase in peak river discharge and
riverine inundation near the outlet, where the highest coastal water levels during Irene lead to elevated maximum discharge
values along the lower Delaware River.

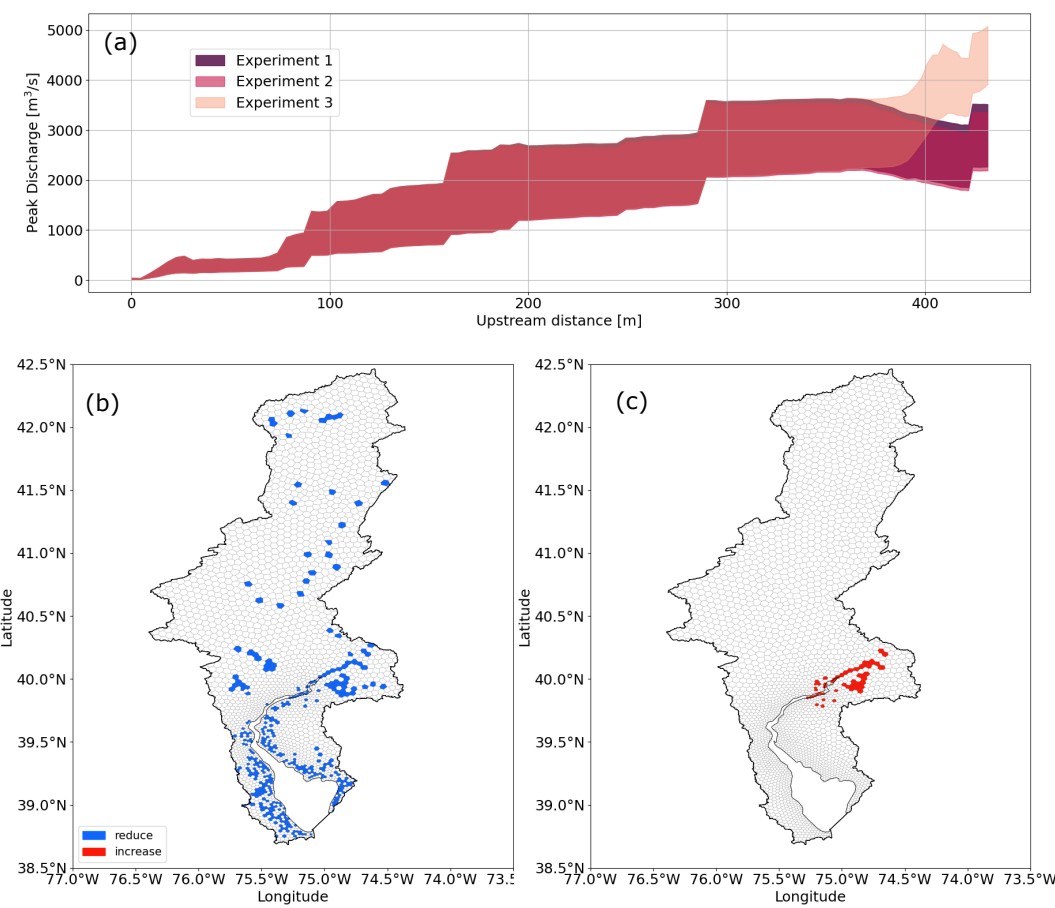

**Figure 6: Comparison of flood impacts of model coupling. (a) Peak discharge along the Delaware River mainstem simulated by 1-way, 2-way land-river and river-ocean coupled simulations in Experiments 1, 2 and 3 (Table 1), respectively. Spatial maps of change in inundation of (b) two-way land river coupled simulations and (c) two-way river-ocean coupled simulations relative to 1-way coupled simulation in Figure 4a. Blue indicates reduced flooded area within the corresponding cell, while red implies an increase in flooded area.**

The impact of the new two-way coupling schemes on accurately capturing the flood extent (Fig. 5) is less significant compared to their effect on modulating the discharge near the river outlet (Fig. 6a), but insightful. Comparing riverine flooding in Experiments 1 and 3, two-way river-ocean coupling improves the flood metrics by 0.01~0.02 and increases FA by $\sim 2.5 \times 10^7$ m$^2$, as a result of a more accurate representation of backwater effects near the river outlet (Fig. 6c). Conversely, the two-way land-river coupling shows a slight reduction in flood metrics and FA, as also indicated in the spatial map (Fig. 6b). The discrepancies observed do not necessarily imply that the inclusion of land-river interactions compromises the results. Rather, they may result from the inherent uncertainties in both data and MOSART simulations, which tend to overestimate riverine flooding. The contrasting behaviors between the two coupling schemes primarily stem from their focus on different spatial and temporal scales. While it is crucial for capturing hydrological processes at larger spatiotemporal scales, the two-way land-river coupling, building upon the macroscale inundation scheme, potentially makes the coupling

less reliable for event-scale riverine flooding. The two-way river-ocean coupling is designed for accurately representing localized interactions between river discharge and tidal or storm surge dynamics that occur at diurnal or semi-diurnal scales. These findings highlight the complex interplay between various coupling approaches and the importance of tailored approaches in flood modeling to address specific hydrodynamic challenges effectively.

## 3.2 Cascading Meteorological Uncertainty

The SEM analysis depicts the possible pathways for the cascading propagation of meteorological and other uncertainties of CF simulations within E3SM (Fig. 7). Specifically, precipitation impacts runoff and infiltration nearly equally but it does not significantly influence soil water storage. The minimal variation in soil water during a TC event is likely because the soil reaches its saturation capacity, especially when rainfall intensity exceeds the soil's infiltration rate. Runoff, which directly contributes to river discharge, positively affects flood simulation in terms of $Q$ and $A_{river}$ in MOSART. Conversely, the impact of infiltration and soil water storage on flooding is negative, as these processes reduce the surface runoff into river channels. Moreover, wind speed combined with air pressure affects sea level variations. The elevated sea level leads to an increase in the coastal inundation area. Additionally, there is a notable interaction between $Q$ and SSH. Increased river discharge tends to elevate local SSH, while high SSH can impede river discharge (Dykstra and Dzwonkowski, 2020). This mutual interaction, frequently observed in CF events, underscores the complexity of the interactive processes influencing both riverine and coastal flooding dynamics, which need to be jointly considered in the two-way river-ocean coupled E3SM.

The cascading of meteorological uncertainty within the E3SM framework is assessed using CV and NMAD (see Eq. 4 and 5) (Fig. 8). Both metrics suggest an amplification of meteorological uncertainty from atmospheric simulations throughout the multi-component system. In the context of riverine flooding, the variability among the ensemble for hydrological drivers such as surface runoff ($Q_{sur}$), subsurface runoff ($Q_{sub}$), and infiltration ($Q_{infl}$) is found to be comparable to that observed in precipitation. However, this variability escalates in riverine flood parameters, i.e., $Q$ and $A_{river}$, where the CV and NMAD values are approximately twofold of those in precipitation. For coastal flooding, uncertainty increases from $U_{wind}$ to SSH, which directly impacts coastal inundation levels ($A_{ocean}$). Much smaller uncertainty is presented in $Q_{soil}$ and $P_{air}$. This analysis highlights the cascading nature of uncertainties from atmospheric inputs through meteorological and hydrological processes to final flood outcomes.

The analysis of the uncertainty path and propagation implies the critical role of hydrological drivers. By quantifying their relative contributions, we can better understand their roles in shaping the variability in riverine flooding outcomes, thereby refining the predictability of ESMs.

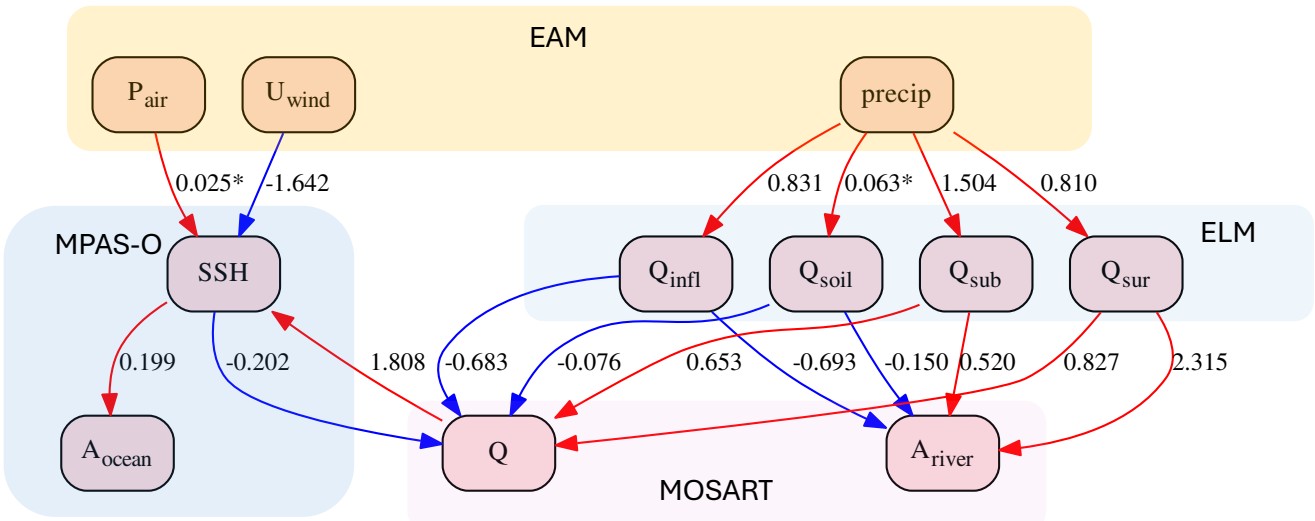

**Figure 7: The structural equation model that describes the influence of variables on their response variables in EAM, ELM, MOSART and MPAS-O. Red and blue arrows show positive and negative influences, respectively. The asterisk sign implies the relationship is not significant with a p-value larger than 0.05.**

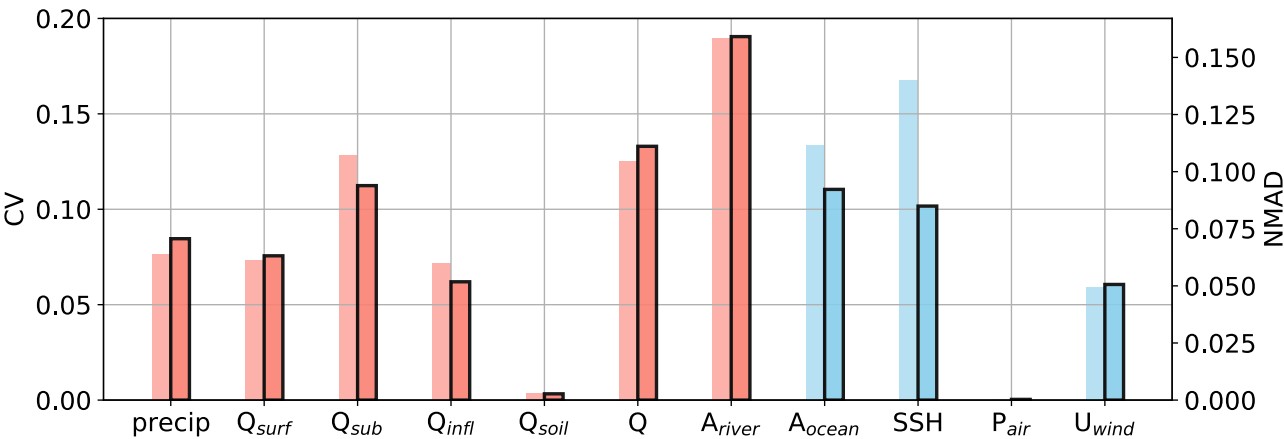

**Figure 8: CV (light bars) and NMAD (dark bars with black margins) computed from the simulation ensembles for the variables selected in Section 2.3, including precipitation ($precip$), surface runoff ($Q_{sur}$), subsurface runoff ($Q_{sub}$), infiltration ($Q_{infl}$) and soil water storage ($Q_{soil}$), river discharge ($Q$), riverine inundation area ($A_{river}$), coastal inundation area ($A_{ocean}$), sea surface height (SSH), air pressure ($P_{air}$) and wind speed ($U_{wind}$). Red and blue bars indicate riverine and coastal flood drivers, respectively.**

## 3.3 Relative Importance of Hydrological Drivers

The extended ensemble simulations provide a wide range of scenarios, encompassing both lower and higher magnitudes of river discharge and riverine inundation compared to those observed during Hurricane Irene (Figure S6 and S7). The ANNs,

trained from the ensemble output, achieve high skill scores. The $r^2$ and NRMSE values for the first ANN are 0.96 and 0.04, respectively, and are 0.97 and 0.03 for the second ANN.

Regarding the cumulative impacts over the entire Irene lifetime, the permutation importance derived from the first ANN highlights the crucial impact of AMC, $f_{drain}$ and $f_{over}$ on $Q_{sur}$, $Q_{sub}$, $Q_{infl}$ and $Q_{soil}$, respectively, whereas precipitation shows more evenly distributed impacts on all the drivers (Fig. 9a). It should be noted that the relatively lower permutation importance values for precipitation do not suggest it is less important compared to the other factors. Rather, this is because in our ensemble, AMC, $f_{drain}$ and $f_{over}$ encompass a broader range of scenarios, whereas precipitation is from the Irene ensemble of simulations that only represent event-specific outcomes. The results of $f_{drain}$ and $f_{over}$ align well with their definitions in ELM (Appendix A), as $f_{drain}$ and $f_{over}$ dominate the change in $Q_{sub}$ and $Q_{sur}$, respectively. Precipitation affects $Q_{sur}$, $Q_{sub}$ and $Q_{infl}$ nearly equally, which corresponds to their similar response presented in Figure 7.

The second ANN analyzes the impact of hydrological drivers on riverine flooding, i.e. river discharge ($Q$) and flooded area ($A_{river}$) (Fig. 9b). Our analysis demonstrates that $Q_{sur}$ and $Q_{sub}$ have similar influences on $Q$, whereas $Q_{infl}$ shows a limited effect. In terms of $A_{river}$, $Q_{sur}$ acts as the dominant factor, whereas $Q_{sub}$ and $Q_{infl}$ are less important but cannot be ignored. $Q_{soil}$ has a minimal impact on both variables. The discrepancy between $Q$ and $A_{river}$ in their responses to these hydrological drivers can be attributed to the nature of the hydrology: river discharge is directly affected by surface and subsurface runoff, which are immediate responses to precipitation. In contrast, inundation across the river basin is more complex, as infiltration exerts a more localized effect and surface runoff may cause rapid flooding in response to intense rainfall. This differential impact implies the need for monitoring day-to-day variations in these drivers throughout the event to understand their dynamic role.

The time evolution of the permutation importance in the second ANN, trained on daily data during Hurricane Irene, illustrates the dynamic roles of hydrological drivers in response to the event and their contributions to riverine flooding. For river discharge, the influence of $Q_{sur}$ and $Q_{sub}$ varies notably before and during the peak flow (Fig. 10). Specifically, peak discharge was observed on August 30 at the river outlet (see Fig 15 in Feng et al., 2024), a period when $Q_{sur}$ was predominant. In contrast, $Q_{sub}$, which typically contributes to baseflow, exerted more influence before the peak. Following the peak, the contributions of $Q_{sur}$, $Q_{sub}$ and $Q_{infl}$ leveled out as significant infiltration into the soil increased soil moisture, revealing a more significant effect of $Q_{soil}$ than that seen in its event-cumulative impact (Fig. 9). The role of soil emerges as vital, acting as a buffer that modulates flooding during the heavy precipitation induced by the TC event. As the event progressed post-peak, there was a noticeable shift with a decreasing impact from $Q_{sub}$ along with a bell-shaped variation in $Q_{infl}$ and $Q_{soil}$. In terms of $A_{river}$, the dynamics slightly differ. $Q_{sur}$ began dominating on August 28, two days earlier compared to $Q$, indicating the routing of discharge from the basin upstream to the outlet. These results reveal the importance of accurate runoff separation in the ESM framework for accurately modeling the time-varying nature of hydrological drivers.

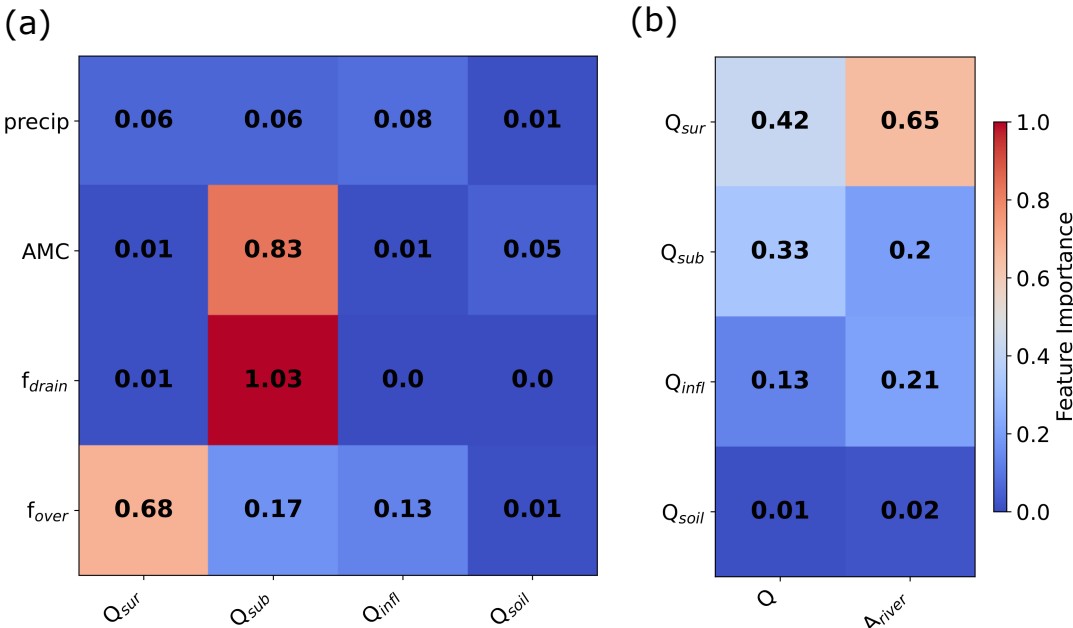

**Figure 9: (a)** Permutation importance of perturbation parameters (precipitation, AMC (antecedent moisture condition), $f_{drain}$ and $f_{over}$) on hydrological drivers of surface runoff ($Q_{sur}$), subsurface runoff ($Q_{sub}$), infiltration ($Q_{infl}$) and soil water storage ($Q_{soil}$). The corresponding box plot of each driver is provided in row 1~4 of Figure S8. **(b)** Permutation importance of hydrological drivers on river discharge ($Q$) and flooded area ($A_{river}$). The coastal inundation area ($A_{ocean}$) is not considered from this analysis as MPAS-O is excluded from the expanded ensemble simulations. The scatter plots of $Q$ and $A_{river}$ against the drivers are respectively provided in the 5th and 6th rows of Figure S6.

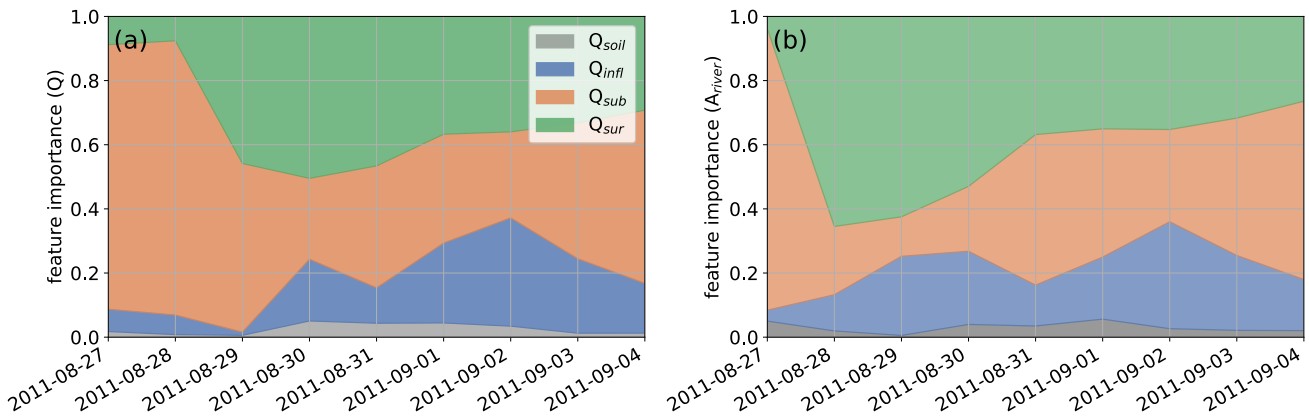

**Figure 10:** Time evolution of permutation importance (scaled between 0 and 1) of the four hydrological drivers for **(a)** river discharge ($Q$) and **(b)** flooded area ($A_{river}$). The corresponding skill scores ($r^2$ and NRMSE) of the ANNs trained using daily data are provided in Figure S9. The Irene-induced peak river discharge is on August 30, 2011.

# 4 Discussions

## 4.1 Uncertainties of CF simulations in E3SM

Integrating different coupling schemes into E3SM has a large impact on the simulated flooding. The exclusion of ocean
coupling resulted in underestimations of the flood extent caused by tide and storm surges, critical for coastal flood assessments. Likewise, we showed that neglecting two-way land-river-ocean interactions distorted the modeled hydrological and hydrodynamic responses to the TC event, as the interactive mechanisms between terrestrial and aquatic systems were overlooked. Therefore, integrating comprehensive coupling mechanisms is essential for improving the predictability of ESMs, particularly in coastal regions vulnerable to complex, multivariate CF events. Additionally, we directly compared the
uncertainty introduced by model coupling with that from atmospheric forcing (Table S1). The atmospheric uncertainty, quantified as the spread of flood metrics across ensemble members in Experiment 3, is comparable to the uncertainty introduced by two-way land-river and river–ocean coupling when only riverine inundation is considered (Exp3 – Exp 1 and Exp 3 – Exp 1). However, when coastal inundation is included, the coupling-induced uncertainty becomes substantially larger across all metrics. The discharge also shows more variability in the river-ocean coupling experiment (Exp 3) than in
the atmospheric ensemble, indicating the significant influence of the two-way river-ocean coupling configuration on flow dynamics. These results highlight the need to consider both meteorological variability and structural model uncertainty when evaluating flood risk in the coupled ESM framework.

The complexities and inherent variabilities of hydrological drivers significantly influence flood exposure through their interactions with meteorological conditions. Particularly, the soil's ability to buffer flood water crucially impacts the onset
and development of floods (Fig. 10) (Blöschl, 2022). Predicting these effects remains challenging, primarily due to the spatial variability of soil characteristics and the spatiotemporal unpredictability of precipitation, such as shifting storm tracks and fluctuating intensity. This uncertainty is further compounded by key hydrological drivers in the land surface model. These parameters affect both the intensity and extent of runoff-driven inundation as well as the soil's response to precipitation (Fig. 9). To address these challenges, CF modeling requires detailed land surface data and advanced modeling
techniques, such as the incorporation of lateral flow (Han et al., 2024) and enhanced land-ocean and land-atmosphere coupling (Lin et al., 2023; Xu et al., 2024), to accurately simulate the interplay between atmospheric, land and river processes.

As discussed above, unlike single-driver flooding that can be simulated in isolated system components, the simulation of CF needs multi-component models, such as E3SM, which are capable of representing the compounding nature among drivers.
However, this also introduces layers of additional uncertainties, particularly in the integration and interaction of model components (Jafarzadegan et al., 2023). Moreover, while regional models often focus on uncertainties arising from prescribed input forcings (Abbaszadeh et al., 2022; Muñoz et al., 2024), the uncertainties in ESMs can propagate bidirectionally through the coupled framework facilitated by two-way coupling schemes, which highlights the contrast in how uncertainties are generated and managed between regional models and ESMs. Quantifying these uncertainties within an

integrated framework is crucial for advancing our understanding of CF but remains a formidable challenge. It necessitates a comprehensive examination of atmospheric, hydrological, oceanic and coupling uncertainties, a task that extends well beyond the capabilities of single-component models.

## 4.2 Definition of "Compound" Flooding

While previous CF studies predominantly focus on the contributions of high discharge, direct runoff, and precipitation to riverine flooding, our analysis reveals the underappreciated roles of other hydrological factors–particularly infiltration and AMC–in the context of CF. These factors significantly influence the flood dynamics in response to TC events. Specifically, we demonstrate that the concurrent occurrence of wet AMC with other CF drivers is not typically accounted for, implying a critical gap in the current CF definition. To capture a broad spectrum of plausible riverine flooding outcomes under varying simulated Irene tracks and AMC conditions, we extracted simulations from the expanded ensemble run by maintaining the default values for $f_{drain}$ and $f_{over}$, resulting in 25 diverse scenarios. These scenarios suggest that a TC preceded by a wet AMC could drastically escalate flood extent. Notably, in all AMC scenarios, we observed a general increase in $Q$ and $A_{river}$ corresponding to increasing precipitation in DRB (Fig. 11a and 11b).

The variability within these simulations shows that the highest discharge was approximately 47% greater than the lowest discharge and 32% higher than during Irene itself (Fig. 11a). Moreover, in the worst-case inundation scenario, flooded areas could increase to more than twice (~2.4) of the flooded areas in the best scenario and the actual Irene event (Fig. 11b). Interestingly, the modeled inundation area for Irene closely aligns with the best-case scenario (Fig. 11b and 11c), despite the fact that Irene occurred under a relatively wet AMC (i.e., 75th percentile AMC). This reflects an asymmetric hydrological response: while drier AMC scenarios show only modest reductions in flood extent, the scenario with saturated soils (AMC100) leads to a disproportionately large increase in peak discharge and inundation. This is likely because, despite the wet soils prior to Irene, there remained sufficient infiltration capacity at the storm's onset. In contrast, further increases in AMC rapidly exceed that capacity, exacerbating surface runoff and flood hazards. This nonlinear amplification highlights the critical role of AMC in modulating compound flood severity. More alarmingly, the expansion of maximum inundation extent from Irene predominantly affects low-lying areas (Fig. 11c), increasing the extent of flooding, raising potential risks to coastal residents, and highlighting the challenges in modeling complex river-ocean interactions, especially considering the effect of sea level rise. These findings suggest a broader definition of CF is needed. Similar to rain-on-snow flooding that may be classified as one type of CF (Zarzycki et al., 2024), a "compounding" event should also consider the co-occurrence of TCs and hydrological extremes, such as AMC, as high AMC can significantly amplify the TC flood impacts.

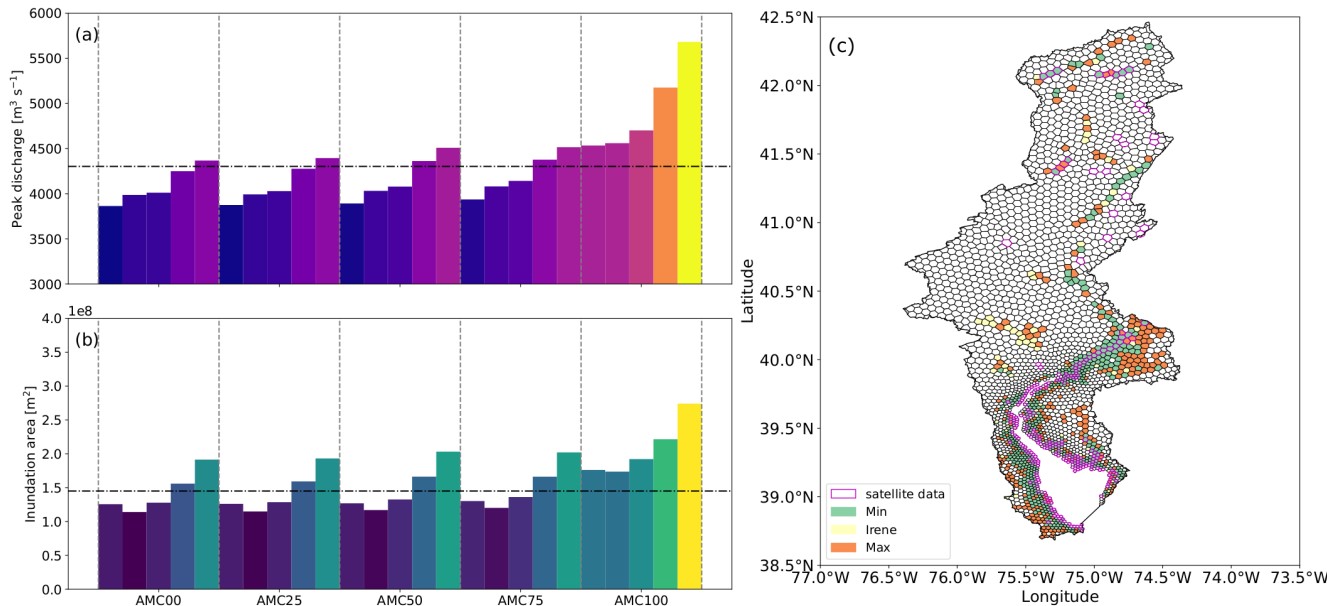

**Figure 11: (a) Peak discharge and (b) riverine inundation area of 25 ensemble simulations. For each AMC, the ensemble runs are sequentially from EAM runs selected in Section 2.4.1. The dashed lines represent the results of the simulation that best describes Irene. (c) The plausible outcomes of inundated extent in DRB with the three colors representing the minimum (best-case scenario), Irene (AMC75) and maximum (worst-case scenario) inundation from the 25 ensemble runs. The purple outline represents the observed flood extent from satellite data, consistent with Figure 4.**

## 4.3 Application of Advanced ESMs in Multivariate Flood Simulations

The application of E3SM in multivariate flood simulations brings a unique set of capabilities, especially when compared to fine-scale regional models. E3SM, with its ability to simulate interactions across various earth system components– atmosphere, land, river and ocean–offers a robust framework for understanding cross-scale environmental dynamics. Even with regional refinement, E3SM may still not be able to provide the street-level details of flood inundation because of missing processes (e.g., pluval inundation) and computational constraints. Although such capability is often crucial for urban

planning and local flood risk management, large-scale E3SM has distinct advantages for broader application scopes. Given the global domain and component complexity, the relatively efficient runtime makes the framework suitable for disentangling interconnected drivers of complex physical processes and their cascading effects through ensemble-based analyses. This efficiency is crucial for running multiple-scenario ensembles, which is essential for understanding the impacts of variability from physical drivers and climate change over extended periods, making it possible to simulate interactions

like the newly developed two-way coupling between land, river and ocean. Although in Section 3.1 our analysis indicates that the land-river two-way coupling has relatively low impacts in short-term modeling of scenarios, its significance could increase in long-term climate simulations where gradual environmental changes play a more prominent role. Furthermore, E3SM provides the potential for climate change simulations, where the interactions of multiple planetary systems need to be considered over global scales and decadal to centennial timescales.

## 4.4 Limitations and Future Work

Despite these strengths, there are inherent challenges and potential sources of uncertainty in using E3SM for flood simulations. These uncertainties can stem from the models' resolution, numerical methods, the accuracy of input data, and the parameterization of complex hydrological and meteorological processes.

One limitation of this study is the exclusion of the ocean model in the expanded ensemble simulations, primarily due to the high computational costs associated with running the global MPAS-O. As a result, the influence of hydrological processes on compound riverine-coastal flooding could not be evaluated using the proposed ANN approach. Future work may focus on enhancing the computational efficiency and feasibility of MPAS-O, for instance, by implement advanced schemes such as local time stepping (Capodaglio & Petersen, 2022; Lilly et al., 2023) and/or developing a regional ocean model within the E3SM framework. Currently, MPAS-O is geared towards global simulations, but adapting it for regional use with the merging of high-resolution local bathymetry data merged could allow for more accurate and locally relevant flood simulations, integrating two-way land-ocean coupling to account for ocean water intrusion (Xu et al., 2024) and its effect on soil moisture along coastlines. This is particularly relevant given our findings on the significant role of soil moisture in the context of TC-induced flooding. As a key driver of coastal flooding, sea level rise (SLR) (Kulp and Strauss, 2019) can also interact with future AMC scenarios (Deb et al, 2023). This interaction may further amplify the flood hazards, which should be considered for more accurate CF risk assessment.

Another avenue for future research involves conducting long-term climate change simulations to assess the impact of climatic drivers on CF dynamics. The existing long-term atmospheric forcing dataset does not adequately capture extreme TC events (Feng et al., 2024). Alternatively, employing a storyline approach (Pettett and Zarzycki, 2023) for event-specific studies could offer a more nuanced and scenario-based method to explore these extreme events and their interactions with other environmental drivers. This approach would not only enhance our understanding of climatic impacts on flooding but also improve the strategic planning and management of flood risks in vulnerable regions.

Our study demonstrates that parameters in runoff generation (i.e., $f_{drain}$ and $f_{over}$) significantly influence river discharge and inundation (Fig. S4 and S5). When these parameters are considered alongside uncertainties in AMC and precipitation, the variability in flood outcomes expands considerably (Fig. S6 and S7). This broader range of variability exceeds that shown in Figure 11, indicating complex interactions between soil properties and hydrological drivers. Given the critical global variability of soil properties, as indicated by the spatial distribution of $f_{drain}$ and $f_{over}$ in Xu et al. (2022a), we anticipate a greater variability in CF impacts that are dependent on soil conditions and land cover (Tran et al., 2024), in addition to topography (Feng et al., 2023b). Furthermore, impervious surfaces, which are prevalent in coastal urban areas, may alter local runoff generation parameters (Zhang et al., 2018). This suggests that these parameters might require high-resolution representation in ELM to accurately reflect their spatial heterogeneity and to better represent urban areas (Li et al. 2024). Future work should focus on refining the spatial resolution in models to better capture the heterogeneity of soil and

urban properties. This improvement could lead to more accurate simulations of how different land surface conditions affect flood dynamics, particularly in diverse geographic settings.

## 7 Conclusions

This study leverages the advanced capabilities of E3SM to improve our understanding of compound river and coastal flooding, highlighting the dynamic interaction between hydrological, riverine and coastal processes. Our research demonstrates that an integrated atmosphere, land, river and ocean system improves the representation of multivariate flooding processes relative to partially coupled configurations, while enabling the analysis of cascading uncertainties through the multi-component Earth system modeling framework. The findings emphasize the significant influence of hydrological

drivers, which can dramatically intensify the impacts of TC-driven flooding. This study not only showcases the robustness of E3SM in bridging gaps in current modeling approaches but also proposes a broader definition of CF that incorporates concurrent hydrological extremes. The implications of our research are profound, advocating for the inclusion of advanced, integrated modeling frameworks in future climate impact assessments to better predict and mitigate the risks of severe flooding events.

**Appendix A: Runoff Generation Parameters in ELM**

This section provides the definitions for the runoff generation parameters $f_{over}$ and $f_{drain}$ in ELM. The fraction of precipitation reaching the ground ($Q_{liq}$) that generates surface runoff ($Q_{sur}$) is determined by the saturation fraction ($f_{sat}$) of the grid cell:

$$Q_{sur} = f_{sat}Q_{liq}, \tag{A1}$$

$$f_{sat} = f_{max}exp\left(-0.5f_{over}z_\nabla\right), \tag{A2}$$

where $f_{max}$ is the potential or maximum saturation fraction of a grid cell, $z_\nabla$ is the water table depth, and $f_{over}$ is a decay factor for surface runoff (Niu et al., 2005). The subsurface runoff is parameterized as an exponential function of $z_\nabla$

$$Q_{sub} = \Theta_{ice}Q_{sub,max}exp\left(-f_{drain}z_\nabla\right), \tag{A3}$$

where $\Theta_{ice}$ is the ice impedance factor, $Q_{sub,max}$ is the maximum drainage rate, and $f_{drain}$ is a decay factor.

**Appendix B: ANN and Permutation Importance**

In our setup, each ANN model included a hidden layer comprising 64 neurons, optimized using an adaptive optimization algorithm, Adam optimizer (Kingma and Ba, 2014). We selected mean square error (MSE) as the loss function to effectively measure the accuracy of predictions during training, which was conducted in the deep learning platform TensorFlow (Abadi et al., 2016). The model completed 600 epochs with a batch size of 32 to ensure thorough learning and convergence. Before

training, the data were randomly split into training and testing datasets, with 80% used for training and 20% for testing, and each variable is normalized with respect to its maxima. The ANN performance was evaluated on the testing dataset using coefficient of determination ($r^2$) and normalized root mean squared error (NRMSE).

Despite the high accuracy achieved by ANN models, it can be challenging to pinpoint the specific influence of individual input variables on output variables (Pires dos Santos et al., 2019). Herein, we employed permutation importance to measure the relative significance of input features within complex ANN models. Permutation importance is a technique used to evaluate the importance of features in a predictive model (Fisher et al., 2019). It assesses the impact of each feature on the model's performance by measuring how much the model's performance decreases when the values of that feature are randomly permuted while leaving other features unchanged (Štrumbelj and Kononenko, 2014; Shrikumar et al., 2017). This method allows quantifying how variations in a single input feature can affect a particular output or overall predictive accuracy. In this study, we computed permutation importance using SHAP (Shapley Additive Explanations, (Lundberg and Lee, 2017)) on the test dataset.

### *Code and data availability.*

The E3SM source code developed in this study is available at the Zenodo repository (Feng, 2024). The HexWatershed simulation results is archived at the Zenodo repository (Liao, 2025). All simulation outputs and processing scripts will be shared upon reasonable request.

### *Author contributions.*

DF and ZT designed the methodology and the numerical experiments. DE, JW, DX, CL, GB and JB prepared the unstructured global meshes, MOSART river networks, model parameter files and EAM ensemble simulations. DF carried out the analysis and the result visualization. DF and ZT wrote the initial draft of the manuscript. All authors contributed to the discussion and review of the results and to the editing of the paper.

### Competing interests

The contact author has declared that none of the authors has any competing interests.

### *Acknowledgement.*

This research has been supported by the Earth System Model Development program areas of the U.S. Department of Energy, Office of Science, Office of Biological and Environmental Research as part of the multi-program, collaborative Integrated Coastal Modeling (ICoM) project (grant no. KP1703110/75415). H.-Y. Li was also supported by the Earth and

Environmental Systems Sciences Division of the U.S. Department of Energy, Office of Science, Office of Biological and Environmental Research as part of the project "A strategic partnership between the College of Engineering at University of Houston and Pacific Northwest National Lab" (grant no. DE-SC0023295). All model simulations were performed using resources available through (a) Research Computing at Pacific Northwest National Laboratory (PNNL) and (b) the National Energy Research Scientific Computing Center (NERSC), a U.S. Department of Energy Office of Science User Facility located at Lawrence Berkeley National Laboratory, operated under Contract No. DE-AC02-05CH11231 using NERSC award BER-ERCAP0027647. PNNL is operated for DOE by Battelle Memorial Institute, United States, under contract DE-AC05-76RL01830.

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
