# Peer review of "Disentangling Atmospheric, Hydrological, and Coupling Uncertainties in Compound Flood Modeling within a Coupled Earth System Model"

_EGUsphere, 2024_

## Referee Comment (RC2)

**Summary**

This article presents a compound flood assessment of Hurricane Irene in the Delaware River Basin. The authors used a new two-way/tightly-coupled approach to quantify the uncertainty of different aspects without the modeling approach, such as the environmental forcing and coupling technique.

**General Comments**

First, the authors need to highlight better the novelty of the paper since it is current version is difficult to identify clearly. For example, they say that their main goal is to assess the uncertainties in the compound flood model, but this has been done before by Muñoz et al. (2024). However, the authors mention this gap (L69-72) but fail to mention the work by Muñoz despite citing them for another reason in the discussion. Another example is their claim that the compound flood needs to be redefined to include AMC and not just TC impacts. However, Bilskie et al. (2021) and Santiago-Collazo et al. (2023) simulated CF from a TC but were preceded by an antecedent rainfall event, which affects the AMC. My point here is that the manuscript needs to go into an intensive literature review process so it can better define the knowledge gaps and cite the appropriate references. In my judgment, presenting the manuscript novelty as an uncertainty assessment of this specific model they used is not enough to impact the science community, as we expect with any manuscript in top-tier journals like this one.

Second, the authors need to improve their introduction section so it can be more comprehensive than its current state. For example, one of the main points of discussion in the manuscript is the coupling technique that their model uses, such as the two-way/tightly coupling approach, but fails to explain to the reader (at least briefly) how this technique works and compares it with others. At a minimum, the authors should briefly define and explain this and cite a reference such as Santiago-Collazo et al. (2019). Similarly, the paragraph (L49-58) that describes the E3Sm model should be moved to methods and include more details of the model itself. Without this, the authors are expecting the reader to go into another reference to read basic details and model configuration, which is not appropriate. At a minimum, they should talk/show their numerical modeling domain, validation results and simulation set, and model assumptions. The authors only cite a paper from Feng et al. (2024) to point the reader to all the important details about the model for this study, which it should not. Furthermore, the author makes various unsupported claims that are not true. For example, (L90) said that model coupling uncertainties have not been studied before (which is not true; see Muñoz et al., 2024) since this coupling technique has been recently developed. However, the authors fail to comment on the many available models that can do this, such as SFICNS, SCHISIM, and the work done by George XU from LSU.

Third, the study limitations should be shown before the results section so the reader is aware of all of them before "believing" the results. For example, I was wondering about the lack of the coastal flood model (deep ocean, for example) since it was not explained in the methods, but then in the limitation section, I learned that the model configuration used in this study does not have one. It is not clear how they model the coastal flooding using their proposed framework since they do not have the coastal flood model (L443) nor use boundary conditions. Thus, I am not even sure if they model coastal processes. I also understand that the authors had a different purpose than to provide a high-resolution model, but the coarse resolution inland is producing an overestimation of the fluvial flood inland. However, the authors said that it is prudent to overestimate the potential flooding from a flood hazard risk assessment perspective (L244-245). As a civil engineer myself, I am in complete disagreement with such a statement. It is not the same to overestimate the flood depth by a couple of inches, but to estimate areas being severely flooded, where there should not be any flood, is just a "bad model." Thus, the author's use of this type of statement represents a lousy justification for their poor validation/calibration processes. I strongly recommend going back to the MOSART model component and calibrating it to a better fit before moving forward with the manuscript. This has a big implication in their study findings since if the simulated storm was a pluvial/fluvial-dominated CF event, then the results would be overestimated greatly, including the uncertainty, since Irene was a coastal flood-dominated event.

**Specific Comments**

- Figure 1: remove the adjectives (e.g., large, heavy, high) for the flood drivers labels. This will make the figure more general and applicable to a broader audience.
- Figure 2: why are the MOSART network lines so different from the real stream network? The authors say that is a limitation but do not offer details on why.
- L227: The word "effectively" should not be used since the authors do not show at least a time series hydrograph, which they are multiple along the river, of its results and the observed to verify how effective the simulation was.
- L 231: does the stream network used in MOSART include the real bathymetry? If so, the data was available?
- L283: there was no discussion/mention of the temporal scale of any model in the manuscript.

---

## Author Comment (AC1)

Response to Reviewers

Title: Disentangling Atmospheric, Hydrological, and Coupling Uncertainties in Compound Flood Modeling within a Coupled Earth System Model

Author Response 1st revision

Reviewer 1
Reviewer Comments:

This paper uses an earth systems modeling framework (1) to simulate compound flooding from Hurricane Irene with different processes represented (e.g., components activated) (2) to investigate the uncertainties that meteorological outputs from an atmospheric model introduce on flooding, and (3) to study the sensitivity of Hurricane Irene flooding under different hydrologic conditions. The authors find that coupling different model components (river, land, ocean) improve the model's ability to generate flooded area from Hurricane Irene (compared to satellite imagery). However, the E3M tends to overestimate flooding overall and requires the use of a computationally expensive local ocean model to resolve the surge inundation. The authors show that the contribution of the uncertainty of the meteorological inputs (via an atmospheric model) is large and that antecedent soil moisture conditions impact the river discharge.

I commend the authors on their analysis and think there are interesting results that inform complexities that arise with modeling hurricane flooding. I particularly enjoyed the analysis of the hydrologic conditions/drivers. The article covers topics relevant to NHESS and could be considered for publication after major revisions. I have included some general and specific comments for the authors.

Author Response:

We appreciate the reviewer for the critical assessment of our work. In the following we address your comments point by point. Our responses and all changes in the revised paper are marked in blue. If you have further questions or concerns, please let us know. Although NHESS does not allow us to share our revised manuscript at this stage of the review process, we provide excerpts throughout the response to help illustrate the changes. We will provide the revised manuscript when invited.

Major concerns:

R1C1:

The gaps are not clearly stated so it is hard to understand the novelty/contribution of this work in the context of existing literature. Overall, this paper seems to focus on how E3Ms new components can be used to study different things including model coupling uncertainty, meteorologic uncertainty, and the impact of different hydrologic conditions on river discharge. These analyses have been done more in-depth in previous studies which the authors could discuss and cite (e.g., Munoz et al. (2022); Gori et al. (2020); Eilander et al. (2023); Bermudez et al. (2021)). Is the focus of the study on improving modeling of compound riverine and coastal flooding compared to regional models (Line 11, 54)? Is it on uncertainties

– model coupling, meteorological, and initial hydrologic conditions (Line 59, 91)? The layout of the introduction could be improved with more relevant citations for the issues the paper is addressing.

Author Response:

Thank you for the insightful comment. We recognize the extensive studies on compound flooding (CF) uncertainty analysis in the literature (Abbaszadeh et al., 2022; Xu et al., 2023; Muñoz et al., 2022; Muñoz et al., 2024), typically based on single or dual-coupled regional/local river and coastal models. While these efforts are highly valued, our research explores a different venue by implementing global Earth System Models (ESMs) with regional refined meshes to simulate complex, multivariate extreme events.

Although ESMs are likely less accurate than regional/local models, they can offer distinct insights about CF due to several prominent features. E3SM, for example, leverages its exascale computing capabilities, provides a tightly coupled framework of multiple Earth system components (i.e., atmosphere, land, river, and ocean) across their global domains, resolves inherent physical processes, and also features interactive coupling among all model components.

Although a recent preprint (Zhang & Yu, 2024) has also demonstrated the importance of using a fully coupled ESM framework for CF modeling, our recent research (Feng et al., 2024) has performed the first integration of new features in E3SM, including two-way land-river-ocean coupling, coastal refined unstructured meshes in atmosphere, land, and river models, and 2-dimensional barotropic ocean models with a wetting and drying scheme. With the ultimate goal of using E3SM to model coastal extremes on a global scale, this study (still running globally) employs a specific TC event and a watershed as a test case. It is important to note that some of these features are relatively new in the domain of ESMs. Alongside model development efforts, it is crucial to evaluate the capability and uncertainty of ESMs in the context of CF modeling. Our developed framework is essential for a comprehensive evaluation of atmospheric, hydrological, and coupling uncertainties.

Uncertainty propagation in fully coupled ESMs remains an important topic in the Earth system modeling community. This complexity is further exacerbated by the two-way land-river-ocean coupling, where downstream models can affect upstream models at high temporal frequencies. Such uncertainty propagation has not been adequately addressed, particularly for extreme events, partly due to the lack of a fully coupled framework that integrates all model components at high resolutions and the high computational cost of running ESM ensembles. Even the study of Muñoz et al. (2024), which inspires our research in its preprint stage, does not comprehensively address atmospheric uncertainty. Although their uncertainty analysis includes a scenario that turned off rainfall, their study relies on a single set of external atmospheric forcing (i.e. ERA5) and does not account for backpropagating uncertainty from two-way coupling. Specifically, we have conducted ensemble atmosphere model simulations to assess the propagation of atmospheric uncertainty through the inherent systems.

Moreover, our study compares atmospheric uncertainty with the coupling uncertainties at both land/river and river/ocean interfaces. While it is true that such coupling techniques have been developed elsewhere, as detailed in our response to R2C2, it remains unclear whether coupling uncertainty is significant compared to atmospheric uncertainty, due to a lack of direct comparisons.

Additionally, we address hydrological uncertainty by running a large ensemble of simulations with perturbed atmospheric forcing and other hydrological parameters, including Antecedent Soil Moisture

Conditions (AMC) and two runoff generation parameters. This large ensemble enables the application of our newly proposed machine learning method and the related SHAP method to understand the relative contributions of hydrological factors. While the importance of AMC has been documented by Deb et al. (2023), none of these studies have demonstrated the role of underlying hydrological factors under different AMC conditions. Bilskie et al. (2021) and Santiago-Collazo et al. (2024) highlighted that the CF impact may be exacerbated if the CF event is preceded by antecedent rainfall, but their studies implemented the rain-on-grid method and did not account for the complex hydrological processes, such as soil moisture, infiltration, or evapotranspiration. We included this discussion in the revision.

With our ensemble run, we discovered that AMC plays a significant role, resulting in more than twice the inundated area if wetter AMC co-occurs with a wetter TC. The methodologies and findings together underscore the novelty of our study, which would not have been possible without the new E3SM framework. More importantly, our research sheds light on future directions and supports the ongoing development of more comprehensive model coupling methods.

In response to this comment, we have restructured the introduction, which begins with a more in-depth literature review on the CF studies, followed by a discussion of recent advancements in ESM model implementation and its associated uncertainties. We elaborated the existing gaps and research questions that need to be addressed. Additionally, we have more explicitly elaborated on the scope, objectives, and novelty of this study.  Please see below the updated introduction starting from the second paragraph.

[revised manuscript text omitted]

R1C2:

The methods section was unclear at times. This work refers to previous papers that describe the modeling framework, validation, and application. However, there is not enough information provided so that the paper is readable by someone in the field. For example, in Section 2.1, are these components 1D or 2D? What are the spatial and temporal resolutions? How have the different model components been used in previous studies and/or validated? By improving the description of the different model components in the paper, it would help readers understand the limitations of the different analysis. For example, if the river model uses a macroscale inundation scheme (Line 284) that should be stated earlier on.

Author Response:

We appreciate the reviewer's suggestion. In the revised manuscript, we have moved the paragraph in the previous introduction (L49-58) that describes E3SM to the methodology section and added more details of the new E3SM framework and the methods developed in our previous studies. The macroscale

inundation scheme is also mentioned in the Methodology section. Please see below the updated Section 2.1.

[revised manuscript text omitted]

R1C3:

The clarity of the manuscript could be improved. The authors should define terms and use them consistently throughout the paper including forcing, drivers, processes, components, responses, conditions, scenarios/experiments (for example Lines 78-81, Lines 210-220, Line 355, Line 371-380, Line 387). The authors should state in the abstract and early in the introduction that E3SM is global. It wasn't until Line 109 that this is mentioned making it unclear in the introduction how/why "regional models" are mentioned in comparison (Line 55-57, Line 391).

Author Response:

Thanks for the comment. We made sure the terminology is defined consistently in the revision. Hydrological drivers represent the surface runoff, subsurface runoff, infiltration and soil water storage, which are defined when they are first mentioned in the methodology section (Section 2.4): "The hydrological drivers we selected for uncertainty analysis include surface runoff, subsurface runoff, infiltration, and soil water storage."

We rephrased "hydrological processes" in the previous L355 and "hydrological parameters" in the previous L383 to "hydrological drivers".

"Forcing" refers to atmospheric forcing in the global E3SM and boundary forcing in regional models.

"Processes" is only used when referring to the general physical processes, such as "atmospheric, land and river processes" in L387.

"components" is typically used for atmosphere, land, river and ocean components in Earth System models (ESMs).

"Response" refers to the change in one variable as a result of a change in another variable. We rephrased "hydrological responses" in the previous L197 and L459 to "hydrological drivers".

"Conditions" refer to a specific state or set of circumstances. For example, "meteorological condition" denotes the current state of the atmosphere, while "soil moisture condition" indicates the current level of moisture content in the soil. We rephrased "various hydrological conditions such as AMC and rainfall scenarios" to "various conditions such as different AMCs and rainfall scenarios".

"Scenario" refers to a set of physical conditions or a simulated situation used to explore the possible outcomes of various conditions. "Experiment" specifically refers to "numerical experiment". We double checked throughout the revised manuscript and made sure they represent their correct definitions.

In the revised introduction, we have added a new third paragraph that includes a brief description of the new E3SM framework. In this paragraph, we emphasized that the simulations were conducted on a global domain.

"Feng et al. (2024) performed the first fully-coupled ESM simulation for CF using the Energy Exascale Earth System Model (E3SM), by integrating recent advancements in E3SM, including regionally refined

unstructured meshes respectively in atmosphere, land/river and ocean components across the entire Earth domain".

R1C4:

The framing of the results and conclusions could be improved. I wonder if it would be more effective to narrow the scope of the manuscript, especially because it addresses so many issues and methods (models, structural equation, ANN). Some of my confusion with the results and discussion relate back to the lack of clarity in the gap/objective of the study. For example, Figure 4. results show that the hit rate is very low when the ocean is not resolved. The large-scale river model is not reliable for event-scale riverine flooding (Line 285). However, discharge and surge are key processes in compound flooding from hurricanes (Lines 35-40) and this is the event the model is being evaluated against.

Your results seem to support the need for resolving flood processes at finer resolutions (e.g., regional or smaller) (again Line 340-342) which E3M is not able to do efficiently due to computational constraints (Line 427). Why did you choose a hurricane to study? Would it be more interesting to look at the impacts of coupling, inputs, conditions, etc. on long-term simulations which the E3Ms seems more suited for (as the authors suggest in Line 435-438)?

Author Response:

We appreciate the suggestion and have revised the introduction to more specifically address the atmospheric, hydrological, and coupling uncertainties within E3SM. We emphasize the novel approach of using global ESMs, which effectively address atmospheric and hydrological uncertainties through a fully coupled framework using simulation ensembles and a detailed land surface model. Additionally, we highlight the importance of understanding how these uncertainties propagate and the related coupling uncertainties. Please refer to our response to R1C1 for the updated introduction.

The use of structural equation modeling (SEM) and ANN depends on the ensemble size. The latter requires a much larger ensemble, which is not feasible for global MPAS-O simulations at the current resolution. We have clarified this point in the revision: "structural equation modeling (SEM) is applied as a path analysis method (Wright, 1921) to trace the flow of data and uncertainty from the 25 ensemble members." In the uncertainty analysis of the hydrological drivers, the expanded ensemble simulation uses the MPAS-O simulated water level as MOSART's coastal boundary condition to reproduce the backwater effects, thereby allowing for the more accurate ANN and SHAP methods. This is mentioned in Section 2.4.

We apologize for any confusion. In L285, we did not mean that the large-scale river model is not reliable for event-scale riverine flooding. Instead, we aim to explain the slightly reduced performance of the two-way land-river coupling in Figure 5. This statement is rephrased to improve clarity: "While it is crucial for capturing hydrological processes at larger spatiotemporal scales, the two-way land-river coupling, building upon the macroscale inundation scheme, potentially makes the coupling less reliable for event-scale riverine flooding".

The low hit rate when the ocean model is excluded is because the river model does not resolve coastal flooding. This was mentioned as "although MOSART is capable of simulating extensive riverine inundation

in coastal regions, it cannot simulate the finer details of inundation immediately adjacent to the coastline (Fig. 4a), where coastal tide and storm surge play a significant role."

In L340-342, we are trying to explain the different behaviors of river discharge $Q$ and riverine inundation extent $A_{river}$. While $Q$ is dominated by surface and subsurface runoff and $A_{river}$ is affected by localized processes such as infiltration. Although a regional or watershed scale model at higher resolutions may improve the flood simulation, our discussion here is not to compare models, which as the reviewer pointed out, is only mentioned in L427. To respond to the reviewer's concern about using a coarser ESM rather than a finer local model to simulate the coastal extremes, we highlighted the motivation in the revised introduction "One significant advantage of ESMs is the integration of multiple earth system components in a single, tightly coupled framework across the global domain. Such a framework allows for predictive understanding of multi-scale flow processes and their interactions with other relevant processes involving heat, energy, biogeochemical and sediment transport, as well as their impacts on Earth's climate (Ward et al., 2020)"

and in the methodology section

"Compared with regional models that may provide more detailed inundation at the street level (Costabile et al., 2023; Ivanov et al., 2021), E3SM excels at coupling processes across various earth system components. This capability is crucial for capturing the complex responses of earth systems to climate change and projecting climate-driven flood hazards."

Regarding the last point of this comment, ESMs are indeed suited for long-term simulations. The MOSART-alone simulation using the same mesh presented good performance in the Mid-Atlantic region over 1991~2019 (Feng et al., 2022). This study, on the other hand, particularly explores the E3SM's capability and uncertainties of simulating coastal extremes using the new configuration as its very first step. The computational constraint posed by the atmosphere model and the fine-resolution global ocean model is another reason why we cannot pursue a long-term simulation in the current stage. The 10-day simulation of MPAS-O alone takes over 6 hours on 640 CPUs. Thus, more computationally efficient schemes, such as local time stepping method (Capodaglio & Petersen, 2022; Lilly et al., 2023), are required to reduce the computational cost. We have added further elaboration to Section 4.4:

"Future work may focus on enhancing the computational efficiency and feasibility of MPAS-O, for instance, by implementing advanced schemes such as local time stepping (Capodaglio & Petersen, 2022; Lilly et al., 2023) and/or developing a regional ocean model within the E3SM framework."

R1C5:

Line 19: The findings in the abstract are not very specific and it would be beneficial to add more detail. For example, what is the measure of the "effectiveness" of the model framework? Is it reproducing the flood extent from Irene?

Author Response:

We appreciate this comment and rephrased "demonstrate E3SM's effectiveness in handling multivariate flooding on the coast" to "demonstrate E3SM's capability in capturing compound flood extent near the coast, with a hit rate over 0.75".

R1C6:

Line 23: The authors state that flood damage could be tripled if Hurricane Irene was preceded by extreme AMC. I do not see this supported anywhere in the results. It would be more appropriate to state the results of changes in the flood hazard (e.g., discharge, flooded area) since these are modeled, not damages.

Author Response:

By "tripled flood damage", we referred to Figure 11b, where the flooded areas in the worst-case scenario could increase to more than twice (~2.4) of the flooded areas in the best scenario and the actual Irene event. We apologize for the confusion and have rephrased this statement to

"For instance, the flooded area could increase by more than twice (~2.4) if Hurricane Irene was preceded by an extreme antecedent soil moisture condition (AMC)."

R1C7:

Line 61: What are the "critical" impacts?

Author Response:

Thanks for the comment. We briefly described the atmospheric forcing impact in the next sentence of the original manuscript (L62): "The water movement in terrestrial and aquatic environments during a TC is strongly influenced by the TC's track and intensity (Pappenberger et al., 2005; Zhong et al., 2010)." In the revision, we rephrased this sentence to improve clarity:

"The river discharge intensity, storm surge levels, and CF inundation extents are directly influenced by the TC's track and intensity, as well as the rainfall rate and timing (Gori et al., 2020a; Pappenberger et al., 2005; Zhong et al., 2010). These factors are the primary drivers of the riverine and coastal flooding dynamics."

R1C8:

Line 110: What is the scale of the coastal refined meshes? Are they comparable size and resolution to "regional models"?

Author Response:

Thanks for the comment. The resolutions of the global EAM, ELM/MOSART and MPAS-O meshes have been added to section 2.1: "The EAM mesh features a global resolution of 100 km, with enhanced refinement to about 25 km over the North Atlantic Ocean. Both ELM and MOSART use a land mesh with a coarse resolution of 60 km globally, which is further refined to 30 km across the contiguous US and to 3 km within the Mid-Atlantic watersheds. The MPAS-O mesh offers the highest resolution of 250 m along the US East Coast, specifically designed to capture estuary dynamics, with a broader global resolution of around 1 km." We note that the resolutions of the global meshes, even regionally refined, could still be lower than regional and local scale models.

R1C9:

Lines 130-135: It would be helpful to include what a perfect score is for each metric (e.g., 0 or 1).

Author Response:

Thank you for your suggestion. In the revised manuscript, We have included a brief explanation for the flood metrics.

"For all metrics, a score of 0 indicates poor performance, while a score of 1 represents perfect model performance."

R1C10:

Line 138: Why did you choose a small threshold of 0.02? I'm assuming meters is the units.

Author Response:

This threshold is unitless, representing the minimum fraction of wet cell. Because we use the binary flood metrics to evaluate the model performance, a cell could be wet or dry even with a very small inundation fraction, especially considering the grid resolution of 5 km. To account for the uncertainty within the satellite data (see our response to R1C21) and the macroscale inundation scheme in the 5-km mesh (see the updated Section 2.1 under the response to R1C2, and our discussion about this uncertainty: "The macroscale inundation scheme may not be sufficient to represent water depth heterogeneity within each cell and the subgrid flow dynamics given the grid resolution of 5 km."), this small threshold is able to eliminate the cell with a very small fraction of inundation potentially caused by the uncertainty.

This threshold is unitless and represents the minimum fraction of a cell considered to be inundated. We have chosen this minimal threshold in order to refine our binary flood metrics that categorize cells as either wet or dry. This approach is particularly pertinent and helps mitigate potential misclassifications in cell status due to the inherent uncertainties presented in the satellite data (see our response to R1C21), the macroscale inundation scheme and the 5-km mesh. The macroscale inundation scheme is now provided in the updated Section 2.1 under the response to R1C2. And its corresponding uncertainty is added to the result discussion: "The macroscale inundation scheme may not be sufficient to represent water depth heterogeneity within each cell and the subgrid flow dynamics given the grid resolution of 5 km." We also elaborated the selection of this threshold in the revision.

"In our simulations, a wet cell is identified if the simulated inundation fraction is above a small unitless threshold of 0.02. This threshold minimizes the influence of cells that may only be marginally inundated—likely due to data and model uncertainties—thus ensuring a more reliable assessment of flood extent."

R1C11:

Line 144-146: Why did you choose a depth of 1m to calculate the inundation fraction?

Author Response:

Thanks for the comment. As we explained in the following sentence, "this number is used to reflect adjustments for the MPAS-O inland bottom elevation bias relative to the actual data". The bathymetry data of the global MPAS-O is sampled from a global gridded bathymetric data set, the General Bathymetric Chart of the Oceans (GEBCO) (IOC and IHO, 2020). This upsampling results in roughly 1-m bias of the inland elevation near the DRB coastline. The inland elevation has to be adjusted to obtain the coastal inundation extent.

In the revision, we made further clarifications here:

"This value represents adjustments made to the MPAS-O inland bottom elevation data near the DRB coastline during the upscale sampling."

In the discussions of the limitation, we highlighted the need for a more accurate global ocean model:

"Currently, MPAS-O is geared towards global simulations, but adapting it for regional use with the merging of high-resolution local bathymetry data could allow for more accurate and locally relevant flood simulations, integrating two-way land-ocean coupling to account for ocean water intrusion (Xu et al., 2024) and its effect on soil moisture along coastlines."

R1C12:

Line 190: I think this should be flood hazards, not flood risks. I would check the use of risk throughout the paper as this is typically defined as the combination of hazard, exposure, and vulnerability.

Author Response:

We acknowledge that "risk" is broadly defined, encompassing flood hazard, exposure, and vulnerability (Kron, 2005). Specifically, flood hazard and exposure risks represent the frequency of flooding events and the extent of human exposure to these events, respectively (Feng et al., 2023). In the revised manuscript, the term "risk" is only used in accordance with its general definition. We checked throughout the manuscript to ensure that the use of "hazard" or "exposure" is consistently accurate and precise. We made the revisions below following this comment.

L119: "CF hazard risks" to "CF hazards"

L191: "flood risks" to "flood exposure"

L246: "flood hazard risk assessment" to "flood exposure risk assessment"

L380 and L407: "flood risks" to "flood exposure"

L414: "increasing risks to coastal residents" to "increasing exposure risks to coastal residents"

R1C13:

Line195: How did you select the 5 ensemble members? How do you define a reasonable spread?

Author Response:

The five ensemble members were chosen from the full set of 25, selected to represent a roughly even distribution of the accumulated discharge and precipitation during Hurricane Irene, which should also result in a wide spread of peak discharge along Delaware River and riverine flood extent (previous Figure S2, now Figure S3). In the revised manuscript, we clarified our selection in the main text:

"selecting 5 ensemble members that represent a roughly even distribution of river discharge and precipitation during Hurricane Irene (Figure S1 and S2)".

Additionally, we have included a new supplementary figure (Figure S2) to further clarify the basis of our selection.

[Figure]

**Figure S2 The accumulated river discharge against precipitation during Irene for the 25 ensemble simulations.**

R1C14:

Lines195-200: What data did you use to get trends in historical soil moisture? To better help the reader understand the range you are looking at, can you state the basin averaged soil moisture range you are using in the simulations? From the supplemental, it looks like the range is 0.065 to 0.087.

Author Response:

Thanks for the comment. The historical soil moisture data is obtained from a historical E3SM land model (ELM) simulation, as stated at L198: "basin-averaged soil moisture during hurricane seasons from 2005 to 2011 as modeled in a historical ELM simulation (Fig. S1)."

We have added more clarification in the caption of Figure S3 (previous Figure S1):

"Time series of soil moisture averaged in Delaware River Basin (DRB) obtained from a historical ELM simulation forced by the Global Soil Wetness Projects version 3 (GSWPv3) climate forcing data set (Kim, 2017) running on the same mesh as Irene simulations."

In the revised main text, we stated the basin averaged soil moisture range as 0.067~0.087 kg m$^{-2}$.

R1C15:

Line 200: Since these two parameters are reported in the results, you should provide more detail on what they represent in the text.

Author Response:

Agree. In the revision, we have clarified:

"Two parameters in ELM ($f_{over}$ and $f_{drain}$), that determine the runoff generation are considered, where $f_{over}$ determines the saturation fraction, i.e. how much surface runoff is generated from precipitation, and $f_{drain}$ controls the subsurface runoff generation (see Appendix A for a detailed definition)."

R1C16:

Line 205: What do you mean by widely distributed impacts?

Author Response:

By "widely distributed impacts", we refer to the selection of each parameter covering a broad range of possible scenarios to quantify their effects. We apologize for the confusion and rephrase this sentence to

"The varying peak discharge observed in the main channel of the Delaware River and the extent of riverine flooding indicate a broad range of possible scenarios captured in simulations that use the perturbed parameters of atmospheric conditions, AMC, and the parameters $f_{over}$ and $f_{drain}$ (Fig. S2 and S4~S6)."

R1C17:

Line 226-237: Some of the statements about the model performance (e.g., "effectively" and "successfully" simulates compound riverine and coastal inundation) might be better suited in the discussion or should be given after you have stated the results that can support these findings.

Author Response:

Agree. We modified the first two sentences to present the capabilities and limitations of the models in a balanced way, focusing on what the results show rather than preemptively describing the models as successful or effective.

"In Experiment 3, the coastal configuration of E3SM employs the coupled MOSART and MPAS-O models to simulate compound riverine and coastal inundation (Fig. 4). The results indicate that MOSART can predict riverine flooding along the lower Delaware River and its upstream tributaries."

R1C18:

Line230-231: Are these inundated cells in the upstream parts of the basin fluvial or pluvial flooding? It is hard to tell if they are located along the river network since that is not shown on Figure 4.

Author Response:

Thanks for the comment. In this study, we do not consider pluvial flooding due to the coarse resolution of the large-scale MOSART model, which is noted in the caption of Figure 4 as 'MOSART simulated riverine inundation. We prefer to use "riverine flooding" over "fluvial flooding", typically because the latter is usually discussed alongside pluvial flooding. In response to this comment, we have updated Figure 4 to include major river channels in the Delaware River Basin (DRB), which are used for MOSART's river network delineation. This addition shows that the inundated areas are primarily along the main channel of the Delaware River.

[Figure]

Figure 4: (a) MOSART simulated riverine inundation (red) against satellite-measured inundation (magenta box). The black dashed box highlights the lower Delaware River reach. (b) E3SM simulated riverine (red) and coastal (blue) total inundation against satellite data (magenta box). The black dashed box represents the coastline of DBE where extensive coastal inundation occurred. In both panels, dark and light colors represent the minimum and maximum inundated extent from the ensemble simulations, respectively. The gray lines are the major river channels.

R1C19:

Line 240: Is the predicted flooded area just an indicator of flooding from the model output or is this measurement used in the comparison with the satellite data?

Author Response:

Thanks for the comment. The predicted flooded area is an indicator of the flood hazard, computed directly from the model's output. The definition has been added to the methodology section (Section 2.1): "The predicted flooded area ($FA$) is calculated by multiplying the flooded fraction by the corresponding cell area." We do not use flooded area for comparison with the satellite data. Due to the uncertainty in satellite data (see our response to R1C21), these data are only used to benchmark the model in terms of the binary metrics.

R1C20:

Figure 4. It would be helpful to spell out hit rate, false rate, and success index in the caption. It would also be helpful to include 1 sentence that lists what the Experiments are again based on the components land, river, ocean.

Author Response:

The caption of Figure 5 has been updated:

"Figure 5: Flood metrics of hit rate ($HR$), false rate ($FR$), success index ($SI$) and flooded area ($FA$) used to compare riverine flooding in Experiments 1~3 and the combined riverine and coastal flooding in Experiment 3. Experiment 1 and 2 include land and river components, while Experiment 3 runs all land, river and ocean components (Table 1). Whiskers extend to 1.5 times the interquartile range from the quartile boundaries."

R1C21:

Line 245: What is the error/uncertainty in the satellite data you are comparing to? Does it perform well in capturing pluvial, fluvial, and coastal flooding from Hurricanes which generally have significant cloud cover? It would be helpful to discuss this in the context of your model results.

Author Response:

We appreciate the reviewer's suggestion and have added more discussions on the uncertainty of the satellite data in the revised section 3.1:

"The overestimation in $FR$ is likely due to the bias in the MODIS satellite data, the macroscale inundation scheme in MOSART and the MOSART mesh resolution. The flood extent dataset (Tellman et al. 2021) is known to result in an underestimation of the actual flooding area due to the cloud cover removing technique (Zhang & Yu, 2024). Its fidelity further decreases in the upstream direction due to the vegetation covers (Sexton et al., 2013)."

R1C22:

Figure 276: Do you mean the influence of the two-way coupling on flooded area (FA) is more significant… not discharge?

Author Response:

We apologize for the confusion. This sentence is rewritten:

"The impact of the new two-way coupling schemes on accurately capturing the flood extent (Fig. 5) is less significant compared to their effect on modulating the discharge near the river outlet (Fig. 6a),"

R1C23:

Line 293-294: Are you saying that the small variations in the soil water storage is because it can't exceed the maximum soil storage capacity? What are the ranges of capacity of the soil?

Author Response:

Yes, the small variation in soil water storage is a result of the soil saturation capability, especially when the rainfall intensity during TCs exceeds the soil's infiltration rate, resulting in water inputs surpass what the soil can absorb. This process depends on a few factors, including the antecedent moisture content of the soil, soil type and rain intensity and duration. In the Delaware River Basin, the soil storage capacity varies significantly across the basin due to a diversity of soil types and their associated physical properties. Soil types range from sandy loams to clay loams. The available water capacity of these soils generally ranges from about 100 mm to 250 mm according to USDA soil surveys. In the response, we elaborate our statement to improve the clarity:

"The minimal variation in soil water during a TC event is likely because the soil reaches its saturation capacity, especially when rainfall intensity exceeds the soil's infiltration rate."

R1C24:

Line 297: Do you consider sea level rise in this modeling framework?

Author Response:

Thanks for the comment. The current modeling framework does not consider sea level rise (SLR) because (a) our simulations do not consider future scenarios and (b) SLR is not yet integrated in MPAS-O, as this functionality is still under development. In the revised discussion, we acknowledged this future research direction:

"As a key driver of coastal flooding, sea level rise (SLR) (Kulp and Strauss, 2019) can also interact with future AMC scenarios (Deb et al, 2023). This interaction may further amplify the flood hazards, which should be considered for more accurate CF risk assessment."

R1C25:

Figure 8: redefine all variables so caption can be understood standalone

Author Response:

The caption of Figure 8 has been updated: "Figure 8: CV (light bars) and NMAD (dark bars with black margins) computed from the simulation ensembles for the variables selected in Section 2.3, including precipitation ($precip$), surface runoff ($Q_{sur}$), subsurface runoff ($Q_{sub}$), infiltration ($Q_{infl}$) and soil water storage ($Q_{soil}$), river discharge ($Q$), riverine inundation area ($A_{river}$),  coastal inundation area ($A_{ocean}$), sea surface height (SSH), air pressure ($P_{air}$) and wind speed ($U_{wind}$). Red and blue bars indicate riverine and coastal flood drivers, respectively."

R1C26:

Figure 10: It would be helpful to have hyetograph and hydrographs of Hurricane Irene, even in the supplemental, to provide context on the results shown in Figure 10.

Author Response:

Thanks for the suggestion. The hyetograph and hydrograph have been added to Supplementary Figure S11.

[Figure]

**Figure S11: Time series of DRB-averaged precipitation and river discharge ($Q$) at the outlet corresponding to Figure 10.**

R1C27:

Does flooded area shown in Figure 5 included the ocean model and then in Figure 9 it is only the river? I would explicitly state this because it is unclear.

Author Response:

Yes, in the expanded ensemble run, the ocean model is excluded as we focus on the hydrological uncertainties specifically. In addition, it is extremely computationally expensive to run the global highresolution MPAS-O for a large ensemble. Thus, as shown in Figure 9, we selected riverine inundation river ($A_{river}$) as the response variable, while in Figure 5, the flooded area (FA) of MOSART and MPAS-O in Experiment 3 included both river and ocean modelled flooded area.

In the revised Figure 5 caption, we explicitly specified that: "MOSART only considers riverine inundation, while MOSART+MPASO accounts for both riverine and coastal inundation."

In the revised Figure 9 caption, we clarified that: "The coastal inundation area ($A_{ocean}$) is not considered from this analysis as MPAS-O is excluded from the expanded ensemble simulations."

R1C28:

Line 355: Did you look at how the time-varying impacts of hydrological processes on compound flooding, not just river discharge?

Author Response:

Thanks for the comment. We exclusively focus on the impact of hydrological processes on riverine flooding in terms of both river discharge ($Q$) and inundation area ($A_{river}$). This is clarified in Section 2.4 (P7L190), "we focus exclusively on riverine flooding in this analysis". The decision was made to "avoid the substantial computational burden associated with MPAS-O". However, to account for the backwater effects on riverine flooding during compound flooding, we did impose the "MPAS-O simulated water level as the coastal boundary condition of MOSART" in the large ensemble simulations (Feng et al., 2022), and in the discussions of limitations (Section 4.4), we acknowledge that "One limitation of this study is the exclusion of the ocean model in the expanded ensemble simulations, primarily due to the high computational costs associated with running the global MPAS-O."

In response to this comment, we further elaborated this point to increase the clarity:

"One limitation of this study is the exclusion of the ocean model in the expanded ensemble simulations, primarily due to the high computational costs associated with running the global MPAS-O. As a result, the influence of hydrological processes on compound riverine-coastal flooding could not be evaluated using the proposed ANN approach."

R1C29:

Figure S6: Where does the 625 number come from?

Author Response:

As described in Section 2.4.1, we selected 5 ensemble members from the atmosphere model runs, 5 AMC scenarios, 5 values for $f_{over}$ and $f_{drain}$, respectively, resulting in a total of 625 ensemble runs. We apologize for being unclear at this point, which has been clarified in the revision in Section 2.4.1: "Using the selected 5 ensemble members, 5 AMC scenarios, 5 values each for $f_{over}$ and $f_{drain}$, we performed a total of 625 (625 = 5 x 5 x 5 x 5) ensemble simulations."

R1C30:

Figure S7: What does the Irene color mean?

Author Response:

Sorry for the typo. The caption of Figure S8 (previous Figure S7) has been corrected to:

"The plausible outcomes of inundated extent in DRB with the five colors representing the minimum, 25th percentile, median, 75th percentile and maximum inundation from the 625 ensemble runs. The magenta boxes represent the flood extent of the satellite data."

Below is the list of newly added references.

[revised manuscript text omitted]

---

## Author Comment (AC2)

Response to Reviewers

Title: Disentangling Atmospheric, Hydrological, and Coupling Uncertainties in Compound Flood Modeling within a Coupled Earth System Model

Author Response 1st revision

Reviewer 2
Reviewer Comments:

This article presents a compound flood assessment of Hurricane Irene in the Delaware River Basin. The authors used a new two-way/tightly-coupled approach to quantify the uncertainty of different aspects without the modeling approach, such as the environmental forcing and coupling technique.

Author Response:

We appreciate the reviewer for the critical assessment of our work. We have carefully addressed the reviewer's suggestions as follows. Excerpts of the revised manuscript are provided to explain our responses to the review comments, and the full revised manuscript will be submitted in the future.

General Comments:

R2C1:

First, the authors need to highlight better the novelty of the paper since it is current version is difficult to identify clearly. For example, they say that their main goal is to assess the uncertainties in the compound flood model, but this has been done before by Muñoz et al. (2024). However, the authors mention this gap (L69-72) but fail to mention the work by Muñoz despite citing them for another reason in the discussion. Another example is their claim that the compound flood needs to be redefined to include AMC and not just TC impacts. However, Bilskie et al. (2021) and Santiago-Collazo et al. (2023) simulated CF from a TC but were preceded by an antecedent rainfall event, which affects the AMC. My point here is that the manuscript needs to go into an intensive literature review process so it can better define the knowledge gaps and cite the appropriate references. In my judgment, presenting the manuscript novelty as an uncertainty assessment of this specific model they used is not enough to impact the science community, as we expect with any manuscript in top-tier journals like this one.

Author Response:

We appreciate the thoughtful comment, which made us realize that the structure of our introduction may have led to the reviewer's assertion that the manuscript's novelty in assessing the uncertainty of this specific modeling framework has limited impact on the scientific community. We recognize the extensive studies on compound flooding (CF) uncertainty analysis in the literature (Abbaszadeh et al., 2022; Xu et al., 2023; Muñoz et al., 2022; Muñoz et al., 2024), typically based on single or dual-coupled regional/local river and coastal models. While these efforts are highly valued, our research explores a different venue

by implementing global-scale regional-refined Earth System Models (ESMs) to simulate complex, multivariate extreme events.

Although ESMs are likely less accurate than regional/local models, they can offer distinct insights about CF due to several prominent features. E3SM, for example, leverages its exascale computing capabilities, provides a tightly coupled framework of multiple Earth system components (i.e., atmosphere, land, river, and ocean) across their global domains, resolves inherent physical processes, and also features interactive coupling among all model components.

Although a recent preprint (Zhang & Yu, 2024) has also demonstrated the importance of using a fully coupled ESM framework for CF modeling, our recent research (Feng et al., 2024) has performed the first integration of new features in E3SM, including two-way land-river-ocean coupling, coastal refined unstructured meshes in atmosphere, land, and river models, and 2-dimensional barotropic ocean models with a wetting and drying scheme. With the ultimate goal of using E3SM to model coastal extremes on a global scale, this study (still running globally) employs a specific TC event and a watershed as a test case. It is important to note that some of these features are relatively new in the domain of ESMs. Alongside model development efforts, it is crucial to evaluate the capability and uncertainty of ESMs in the context of CF modeling. Our developed framework is essential for a comprehensive evaluation of atmospheric, hydrological, and coupling uncertainties.

Uncertainty propagation in fully coupled ESMs remains an important topic in the Earth system modeling community. This complexity is further exacerbated by the two-way land-river-ocean coupling, where downstream models can affect upstream models at high temporal frequencies. Such uncertainty propagation has not been adequately addressed, particularly for extreme events, partly due to the lack of a fully coupled framework that integrates all model components at high resolutions and the high computational cost of running ESM ensembles. Even the study of Muñoz et al. (2024), which inspires our research in its preprint stage, does not comprehensively address atmospheric uncertainty. Although their uncertainty analysis includes a scenario that turned off rainfall, their study relies on a single set of external atmospheric forcing (i.e. ERA5) and does not account for backpropagating uncertainty from two-way coupling. Specifically, we have conducted ensemble atmosphere model simulations to assess the propagation of atmospheric uncertainty through the inherent systems.

Moreover, our study compares atmospheric uncertainty with the coupling uncertainties at both land/river and river/ocean interfaces. While it is true that such coupling techniques have been developed elsewhere, as detailed in R2C2, it remains unclear whether coupling uncertainty is significant compared to atmospheric uncertainty, due to a lack of direct comparisons.

Additionally, we address hydrological uncertainty by running a large ensemble of simulations with perturbed atmospheric forcing and other hydrological parameters, including Antecedent Soil Moisture Conditions (AMC) and two runoff generation parameters. This large ensemble enables the application of our newly proposed machine learning method and the related SHAP method to understand the relative contributions of hydrological factors. While the importance of AMC has been documented by Deb et al. (2023), none of these studies have demonstrated the role of underlying hydrological factors under different AMC conditions. Bilskie et al. (2021) and Santiago-Collazo et al. (2024) highlighted that the CF impact may be exacerbated if the CF event is preceded by antecedent rainfall, but their studies implemented the rain-on-grid method and did not account for the complex hydrological processes, such as soil moisture, infiltration, or evapotranspiration. We included this discussion in the revision.

With our ensemble run, we discovered that AMC plays a significant role, resulting in more than twice the inundated area if wetter AMC co-occurs with a wetter TC. The methodologies and findings together underscore the novelty of our study, which would not have been possible without the new E3SM framework. More importantly, our research sheds light on future directions and supports the ongoing development of more comprehensive model coupling methods.

In response to this comment, we have restructured the introduction, which begins with a more in-depth literature review on the CF studies, followed by a discussion of recent advancements in ESM model implementation and its associated uncertainties. We elaborated the existing gaps and research questions that need to be addressed. Additionally, we have more explicitly elaborated on the scope, objectives, and novelty of this study.  Please see below the updated introduction starting from the second paragraph.

[revised manuscript text omitted]

R2C2:

Second, the authors need to improve their introduction section so it can be more comprehensive than its current state. For example, one of the main points of discussion in the manuscript is the coupling technique that their model uses, such as the twoway/tightly coupling approach, but fails to explain to the reader (at least briefly) how this technique works and compares it with others. At a minimum, the authors should briefly define and explain this and cite a reference such as Santiago-Collazo et al. (2019). Similarly, the paragraph (L49-58) that describes the E3Sm model should be moved to methods and include more details of the model itself. Without this, the authors are expecting the reader to go into another reference to read basic details and model configuration, which is not appropriate. At a minimum, they should talk/show their numerical modeling domain, validation results and simulation set, and model assumptions. The authors only cite a paper from Feng et al. (2024) to point the reader to all the important details about the model for this study, which it should not. Furthermore, the author makes various unsupported claims that are not true. For example, (L90) said that model coupling uncertainties have not been studied before (which is not true; see Muñoz et al., 2024) since this coupling technique has been recently developed. However, the authors fail to comment on the many available models that can do this, such as SFICNS, SCHISIM, and the work done by George XU from LSU.

Author Response:

Thank you for the feedback. We are aware of the comprehensive discussion on the different model coupling approaches reviewed by Santiago-Collazo et al. (2019). We cited the reference and only briefly mentioned the limitation of the one-way coupling method on compound flooding: "Traditional modeling

approaches that rely on one-way coupling between any two model components thus have a limited ability to capture CF (Santiago-Collazo et al., 2019)", as we consider their classification between two-way loosely coupling and tightly coupling is slightly different than the definition in Earth System Models (ESMs) that consider multiple components as a single modeling system. In the revision, we have further elaborated on the difference between one-way and two-way coupling:

"The cascading uncertainty in ESMs becomes even more complex with two-way interactive model coupling. In online two-way coupling, a downstream model, while receiving data from its upstream component, sends back real-time computed information at predefined time intervals, enabling bi-directional data exchange. For instance, a river model may send floodplain inundated water extent to the land surface model for estimating flood water infiltration on the floodplain (Xu et al., 2022b). Similarly, an ocean model provides its predicted water levels (Bao et al., 2022; Bao et al., 2024; Feng et al., 2024), velocities (Zhang et al., 2024) or fluxes (Shen et al., 2024) to the river model for capturing the backwater effect."

We apologize for any confusion caused, leading the reviewer to believe we made unsupported false claims about model coupling and the associated uncertainty. While some of our ideas on addressing uncertainty propagation in a coupled modeling system were inspired by Muñoz et al., 2024, their focused uncertainties using one-way coupled regional models differ from our study's focus and objectives. Specifically, our aim is to evaluate the capability and uncertainty of ESMs in simulating complex CF events. We concentrate on a different set of uncertainties inherent to the Earth modeling system, which are assessed through ensemble atmospheric simulations. In contrast, regional models are more susceptible to boundary forcing uncertainties, which are not applicable to our framework, as the coupled framework in the global domain does not require boundary forcing.

Additionally, other two-way coupled regional modeling studies (Bao et al., 2022; Bao et al., 2024; Zhang et al., 2024) primarily couple regional river and ocean models, such as WRF-Hydro, SCHISM, and ROMS. These model development studies lack the in-depth uncertainty analysis as done by Muñoz et al. (2024). The question of whether coupling uncertainty is significant, given the dominance of atmospheric uncertainty in such cases, remains unaddressed. We appreciate the reviewer's valuable recommendations and have expanded our literature review accordingly. Please see the updated introduction in our response to R2C1.

Following the reviewer's suggestion, we have relocated the detailed description of E3SM to the methodology section. We have intentionally not duplicated the mesh and result presentation in Feng et al., (2024). However, in the revised manuscript, we carefully describe the methodology being employed here, including the modeling schemes, mesh resolution, spatial and temporal scales and validation metrics. Please see below the updated Section 2.1.

[revised manuscript text omitted]

R2C3:

Third, the study limitations should be shown before the results section so the reader is aware of all of them before "believing" the results. For example, I was wondering about the lack of the coastal flood model (deep ocean, for example) since it was not explained in the methods, but then in the limitation section, I learned that the model configuration used in this study does not have one. It is not clear how they model the coastal flooding using their proposed framework since they do not have the coastal flood

model (L443) nor use boundary conditions. Thus, I am not even sure if they model coastal processes. I also understand that the authors had a different purpose than to provide a high-resolution model, but the coarse resolution inland is producing an overestimation of the fluvial flood inland.

However, the authors said that it is prudent to overestimate the potential flooding from a flood hazard risk assessment perspective (L244-245). As a civil engineer myself, I am in complete disagreement with such a statement. It is not the same to overestimate the flood depth by a couple of inches, but to estimate areas being severely flooded, where there should not be any flood, is just a "bad model." Thus, the author's use of this type of statement represents a lousy justification for their poor validation/calibration processes. I strongly recommend going back to the MOSART model component and calibrating it to a better fit before moving forward with the manuscript. This has a big implication in their study findings since if the simulated storm was a pluvial/fluvial-dominated CF event, then the results would be overestimated greatly, including the uncertainty, since Irene was a coastal flood-dominated event.

Author Response:

Our framework indeed has the global ocean model MPAS-O integrated and run in the 25 ensemble simulations, which is described in Section 2.1. The ocean model is only excluded from the expanded ensemble simulations as explained in Section 2.4, in which case MPAS-O simulated water level provides the coastal boundary condition to MOSART (L190-193). In the revision, we included more details about the MPAS-O setup and validation. Please see our response to R2C2 for the updated Section 2.1.

MOSART has been validated against the measured streamflow at USGS gauges along the Delaware River main channel with averaged coefficient of determination ($r^2$) of 0.79 and Kling–Gupta efficiency (KGE) (Gupta et al., 2009) of 0.84, which is comparable to that of a high-resolution regional model using the same atmospheric forcing (Deb et al., 2024). The bias in the streamflow simulation is due to the EAM simulation of Hurricane Irene, which does not have nudging nor is its output bias-corrected in the ensemble simulation. The overestimation in FR is likely due to the bias in the MODIS satellite data, the macroscale inundation scheme in MOSART, and the 5-km resolution of the MOSART mesh. The flood extent dataset (Tellman et al. 2021) is known to likely underestimate the actual flooding area due to the uncertainty in the cloud cover removing technique, which is also reported in Zhang & Yu, 2024. Its fidelity further decreases in the upstream direction due to vegetation cover (Sexton et al., 2013). Such data, unlike high-resolution lidar data, should be only used to "benchmark" (rather than "validate") the modeled results. This was mentioned in the previous L128. The macroscale inundation scheme typically used in large-scale river models assumes no between-cell flux exchange and estimates the inundation fraction using the elevation profiles within the grid cell (Luo et al., 2017; Yamazaki et al., 2011). Such a scheme may not be sufficient to represent water depth heterogeneity within each cell and the subgrid flow dynamics given the grid resolution of 5 km. We agree with the reviewer that high-resolution local-scale models, which are able to provide street level flood prediction, consider the bias of a few inches as poor performance. However, global-scale Earth System Models (ESMs) offer their unique advantages in simulating coastal extremes, which is further elaborated in the updated introduction: "One significant advantage of ESMs is the integration of multiple earth system components in a single, tightly coupled framework across the global domain. Such a framework allows for predictive understanding of multi-scale flow processes and their interactions with other relevant processes involving heat, energy, biogeochemical and sediment transport, as well as their impacts on Earth's climate (Ward et al., 2020)." See our response to R2C1 for the updated introduction.

We agree with the reviewer that it is not appropriate to state the prudence of overestimating the potential flooding. In response, we first clarified the objective and advantage of the global E3SM in the introduction. We then added more details about the MOSART macroscale inundation scheme and the validation result in Section 2.1. Finally, in the revised Section 3.1, we provided more elaboration on this bias: "The overestimation in $FR$ is likely due to the bias in the MODIS satellite data, the macroscale inundation scheme in MOSART, and the MOSART mesh resolution. The flood extent dataset (Tellman et al. 2021) could underestimate the actual flooding area due to the uncertainty in the cloud cover removing technique (Zhang & Yu, 2024). Its fidelity further decreases towards upstream due to the existence of vegetation covers (Sexton et al., 2013). In addition, the macroscale inundation scheme may not capture the subgrid connectivity given the grid resolution of 5 km (Xu et al., 2022b)."

Specific Comments:

R2C4:

Figure 1: remove the adjectives (e.g., large, heavy, high) for the flood drivers labels. This will make the figure more general and applicable to a broader audience.

Author Response:

Thanks. Figure 1 has been updated following the reviewer's suggestion.

[Figure]

**Figure 1: Compound flooding processes in a coastal river basin during a TC event. This conceptual diagram shows the key elements contributing to CF, resulting from combined riverine and coastal inundation along the river channels and adjacent coastal areas.**

R2C5:

Figure 2: why are the MOSART network lines so different from the real stream network? The authors say that is a limitation but do not offer details on why.

Author Response:

We appreciate this comment. Upon checking the MOSART river network and the Delaware River streams in Figure 2b, we don't think they are very different at the 5-km resolution. In particular, we employed a rigorous way to generate the river network using HexWatershed (Liao et al., 2022, 2023a, b), a watershed and flow direction model that supports unstructured meshes for river routing models. HexWatershed employs a topology-based method to delineate river networks in the mid-Atlantic region (Lehner et al., 2008). This model generates flow direction across the domain using a combination of depression filling and stream burning algorithms. These algorithms are designed to eliminate local depressions with minimal alterations to the surface elevation, thereby producing essential flow routing parameters such as flow direction maps, channel slopes, and drainage areas. This information is complemented in the updated Section 2.1. See our response to R2C2 for the new section.

We agree with the reviewer that the original depiction of Figure 2b might lead to confusion, particularly due to the prominently displayed subbasin (HUC8) boundaries in thick black lines. Additionally, it's important to note that the Delaware streams shown in the figure, sourced directly from the Delaware River Basin Commission, were intended solely to illustrate geographic conditions and not for generating the river network. In the updated Figure 2b, we have omitted the Delaware streams and the MOSART river network. Instead, we now directly showcase the major channels of the Delaware River as derived from the HexWatershed output. We also changed to the subbasin boundaries to dashed gray lines.

[Figure]

**Figure 2: (a) The multi-component E3SM framework and drivers used for analyses within each model component. The black arrows represent the data flow via the one-way coupled framework. The white arrows are the new flow directions from the 2-way land-river and river-ocean models. (b) Map of Delaware river basin (DRB), Delaware bay estuary (DBE), and the observed (red) and modeled (black) Irene tracks. The topographic map in (b) is from the ESRI world topographic map (ESRI, 2012).**

R2C6:

L227: The word "effectively" should not be used since the authors do not show at least a time series hydrograph, which they are multiple along the river, of its results and the observed to verify how effective the simulation was.

Author Response:

The validation metrics of MOSART and MPAS-O against observations are now provided in Section 2.1. We prefer not to show the hydrograph time series as it has already been presented in Figure 5 of Feng et al., (2024). Additionally, we have modified the first two sentences of this paragraph to present the capabilities and limitations of the models in a balanced way, focusing on what the results show rather than preemptively describing the models as successful or effective.

"In Experiment 3, the E3SM coastal configuration of E3SM employs the coupled MOSART and MPAS-O models to simulate compound riverine and coastal inundation (Fig. 4). The results indicate that MOSART can predict riverine flooding along the lower Delaware River and its upstream tributaries."

R2C7:

L 231: does the stream network used in MOSART include the real bathymetry? If so, the data was available?

Author Response:

The delineation of the global river network and the riverbed slope of MOSART uses the 90-m HydroSHEDS digital elevation model (DEM) (Lehner et al., 2008). The river geometry parameters of bankfull width and depth were derived using the power law function with bankfull discharge (Andreadis et al., 2013) as there is no river bathymetry available for the whole watershed or globally. This information has been added to Section 2.1.

R2C8:

L283: there was no discussion/mention of the temporal scale of any model in the manuscript.

Author Response:

Sorry for the confusion. In the revision, we have clarified the model's output frequency and coupling frequency.

[revised manuscript text omitted]

---

## Author Response (AR2)

Response to Reviewers

Title:  Disentangling Atmospheric, Hydrological, and Coupling Uncertainties in Compound Flood Modeling within a Coupled Earth System Model

Author Response 2nd revision

Editor comments:

My decision is that the paper can be published after minor revisions. This is based on multiple rounds of review and reports by three reviewers. Currently, one reviewer is satisfied with the revisions, but the other reviewer (who did not review the first version of the manuscript) has a few major comments that need to be addressed. The authors should address all comments from Reviewer #2 before resubmitting the manuscript. Specifically, they should expand and improve the literature review related to uncertainty analysis methods. Second, they should improve the presentation of the results by comparing the relative uncertainties from different sources.

Author Response:

We would like to sincerely thank the editor and reviewers for their valuable comments and recommendations. We have carefully addressed the reviewer's suggestions as follows.

Reviewer 2
Reviewer Comments:

This manuscript applies an earth system model, E3SM, to study the coupling and meteorological uncertainties associated with compound flood modeling. The model is applied to simulate inundation in the Delaware River Basin and Estuary during Hurricane Irene.

Author Response:

We appreciate the reviewer for the critical assessment of our work. In the following we address your comments point by point. Our responses and all changes in the revised paper are marked in blue.

Major comments:

R2C1:

Given that the focus of this paper is on assessing sources of uncertainty in the coupled compound flood framework, the literature review should be expanded to discuss the approaches used for uncertainty analysis in previous studies and their pros/cons. This would help to justify the choice of SEM and ANN for this study.

Author Response:

Thank you for the insightful comment. We agree that a more comprehensive review of uncertainty quantification methods would benefit the manuscript and help clarify the rationale behind our approach.

We would also like to emphasize that our focus is on uncertainty quantification within a fully coupled Earth system modeling framework, which remains an underexplored area in compound flood research. To assess coupling uncertainty, we conducted controlled experiments (as shown in Table 1), comparing simulated discharge and inundation under different coupling configurations, all driven by the same atmospheric ensemble. For cascading meteorological uncertainty, we applied two complementary methods: (1) ensemble spread metrics (NMAD and CV), and (2) Structural Equation Modeling (SEM), which reveals interdependencies among atmospheric, hydrological, and oceanic variables. SEM is particularly well-suited for identifying cause-effect relationships in complex systems with multiple interacting drivers. Due to the high computational cost of fully coupled simulations with the global ocean model activated, we did not apply ANN-based methods to the meteorological uncertainty analysis. However, we were able to expand the ensemble size for hydrological uncertainty analysis, which enabled us to use ANN to examine the relative influence of hydrological drivers on discharge and inundation. This follows a similar approach to Muñoz et al. (2024), who used random forests to identify dominant drivers, although our ANN model allows for multivariate predictions (discharge and inundation).

In the revised manuscript, we have expanded the introduction (L118~L133) to include a brief review of existing methods for uncertainty analysis, such as ensemble modeling, statistical modeling, machine learning approaches and SEM, along with their advantages and limitations.

"The above-mentioned uncertainties are complicated but must be carefully evaluated for ESMs as they will be more frequently applied for CF simulations in the context of climate change. A variety of approaches have been adopted for understanding the uncertainties of CF modeling. These approaches

offer trade-offs between computational cost, physical interpretability, and the ability to disentangle complex drivers. Ensemble-based methods remain a primary strategy for characterizing the cascading uncertainty from the forcing data (Hamill et al., 2011; Hou et al., 2017; Villarini et al., 2019). Multiple realizations with perturbed initial conditions and/or model physics represent a range of scenarios that evolve differently based on the dynamics of the models (Blanton et al., 2020; Nederhoff et al., 2024; Saleh et al., 2017; Wang et al., 2024). Probabilistic frameworks, such as Bayesian inference (Beven and Binley, 1992), provide more robust treatment of parameters and model uncertainties (Naseri & Hummel, 2022), but often rely on strong assumptions and intensive sampling. Machine learning techniques have been increasingly applied to flood modeling (Hu et al., 2019) and are effective at capturing nonlinear relationships of CF drivers (Muñoz et al., 2024), though they require large training datasets and may sacrifice physical interpretability (Shen et al., 2023). Structural equation modeling (SEM; Wright, 1921) has also been adopted to disentangle complex, interacting processes (Du et al., 2015; Santoro et al., 2023). SEM offers a balance between statistical rigor and interpretability in multi-driver systems without a significantly amount of data. Despite these advances, uncertainty quantification within fully coupled ESM frameworks remains relatively underexplored due to high computational demands and limited methodological integration across domains."

R2C2:

The manuscript could be improved by better highlighting and comparing the relative uncertainties from different sources. Different methods were used to quantify the coupling uncertainty, the meteorological uncertainty, and the uncertainty from hydrological driver propagation, and I don't see a definitive comparison between or synthesis of the results. Line 438-442 states that "uncertainty from the atmosphere simulations is comparable to that of two-way river-ocean coupling… but is considerably smaller than the uncertainty of two-way river-ocean coupling if the MPAS-O modeled inundation is excluded." However, it is unclear to me where the "uncertainty from the atmosphere simulations" is clearly reported and how the magnitudes were compared.

Author Response:

We appreciate the reviewer's suggestion and acknowledge that the comparison between different sources of uncertainty, particularly between atmospheric forcing and model coupling, was not clearly presented in the original manuscript. While the cascading meteorological uncertainty and the hydrological uncertainty target different aspects of the system and are therefore not directly comparable (the former concerns the propagation of uncertainty through interconnected system drivers, while the latter focuses on the influence from distinct hydrological drivers), we agree that a more explicit comparison between the atmospheric forcing uncertainty and coupling uncertainty is necessary.

To clarify, the standard deviation values reported in lines 438–442 of the previous manuscript (e.g., $\sigma = 0.015$ for HR, 0.014 for FR, etc., in Experiment 3) represent the variability in flood metrics across all ensemble members of atmospheric forcing, thereby quantifying the uncertainty introduced by atmospheric variability. In contrast, coupling uncertainty can be assessed by comparing the flood metrics across Experiments 1, 2, and 3 using the mean values of the ensemble simulation.

In response to this comment, we have added a new Supplementary Table S1 that directly compares these two sources of uncertainty. This table reports: (1) the magnitude of differences in flood metrics between Experiments 1, 2, and 3 (reflecting coupling uncertainty) and (2) the standard deviation of flood metrics across all ensemble members (reflecting atmospheric uncertainty). Additionally, we have revised the discussion section in the main text to elaborate on this comparison (L461-L469).

"Additionally, we directly compared the uncertainty introduced by model coupling with that from atmospheric forcing (Table S1). The atmospheric uncertainty, quantified as the spread of flood metrics across ensemble members in Experiment 3, is comparable to the uncertainty introduced by two-way land-river and river–ocean coupling when only riverine inundation is considered (Exp3 – Exp 1 and Exp 3 – Exp 1). However, when coastal inundation is included, the coupling-induced uncertainty becomes substantially larger across all metrics. The discharge also shows more variability in the river-ocean coupling experiment (Exp 3) than in the atmospheric ensemble, indicating the significant influence of the two-way river-ocean coupling configuration on flow dynamics. These results highlight the need to consider both meteorological variability and structural model uncertainty when evaluating flood risk in the coupled ESM framework."

**Table S1 Comparison of uncertainty in flood and discharge metrics due to model coupling and atmospheric forcing. Coupling-induced uncertainty is represented by the difference in the metrics between Experiments 2 or 3 and Experiment 1, averaged across ensemble simulations. Atmospheric uncertainty is quantified as the standard deviation of metrics across ensemble members in Experiment 3.**

| Uncertainty source | HR | FR | SI | FA [$\times 10^7 m^2$] | Q [$m^3/s$] |
|---|---|---|---|---|---|
| Two-way land-river coupling (riverine flooding) (Exp 2 – Exp 1) | -0.076 | 0.007 | -0.020 | -6.186 | -124.342 |
| Two-way river-ocean coupling (riverine flooding) (Exp 3 – Exp 1) | 0.015 | -0.004 | 0.005 | 2.480 | 1641.695 |
| Two-way river-ocean coupling (riverine&coastal flooding) (Exp 3 – Exp 1) | 0.346 | -0.098 | 0.129 | 54.185 | |
| Atmospheric forcing (standard deviation in Exp 3) | 0.015 | 0.014 | 0.01 | 8.500 | 326.094 |

Other Comments:

R2C3:

Line 179-180: Is the elevation adjustment of 1 meter spatially uniform? Is this justified by the data?

Author Response:

We apologize for the confusion. To clarify, the 1-meter threshold for determining inundation in MPAS-O cells is not a spatially uniform adjustment of elevation data itself, nor does it reflect a deficiency in the underlying GEBCO bathymetry data. Instead, it is a practical criterion introduced during post-processing to reduce minor inundation signals that can appear when aggregating from the 250-m MPAS-O mesh to the coarser MOSART grid. Therefore, this 1-meter threshold is justified not by the original elevation data, but rather by the resolution gap and the need to maintain consistency in identifying meaningful inundation extents when aggregating results across different spatial scales. We will add a brief clarification in the manuscript to reflect this point explicitly (L197~L200).

"Within each MOSART grid cell, the inundation fraction is determined by the percentage of MPAS-O cells with a simulated water depth over 1 m. This threshold does not imply a spatially uniform adjustment of the GEBCO bathymetry data used by MPAS-O. Instead, it serves as a practical criterion to mitigate biases arising from upscaling inundation extents from the higher-resolution MPAS-O mesh to the coarser MOSART grid."

R2C4:

Line 198-199: Please clarify here that Experiment 1 used one-way coupling while Experiment 2 used two-way coupling.

Author Response:

Thanks. This has been clarified in the revision (L216~L217):

"The first two experiments implement one-way and two-way coupled ELM and MOSART, respectively, while the third experiment interactively couples MPAS-O with MOSART."

R2C5:

Line 248-249: By "roughly even distribution", do the authors mean that the discharge and precipitation values are sampled at even intervals across the range of values modeled, or that the values are applied over the study domain in an event spatial pattern? Please clarify.

Author Response:

Sorry for the confusion. We confirm that by "roughly even distribution," we mean that the selected ensemble members were chosen such that their simulated discharge and precipitation span the full range of modeled values, with values approximately evenly spaced across that range. We have revised the manuscript to clarify this point (L266~L267).

"The original EAM ensemble is expanded by first selecting 5 ensemble members whose river discharge and precipitation values span the full range of the ensemble and are approximately evenly spaced across that range during Hurricane Irene"

R2C6:

Line 264-265: It was not clear what data was used to train/test the ANN. Are the 625 ensemble simulations from the coupled model used? What was the split for testing and training?

Author Response:

Thanks. The input and output variables shown in Figure 3 were derived from the full set of 625 ensemble simulations. These simulations provided the dataset used for both training and testing the two-stage ANN. As now clarified in Appendix B, the dataset was randomly split into 80% for training and 20% for testing.

Revised Main Text: "To quantify the relative importance of each hydrological driver of CF, we employed a two-stage Artificial Neural Network (ANN) approach (Fig. 3), trained and tested using data from all 625 ensemble simulations."

Revised Appendix B: "Before training, the data were randomly split into training and testing datasets, with 80% used for training and 20% for testing,"

R2C7:

Line 290-294 and Fig 4: In panel (a), which only uses MOSART, it is not clear to me why the cells immediately adjacent to the river and bay are not inundated but the adjacent inland cells are. It seems that if the flood is propagating from the river into the floodplain, the shoreline cells should also be inundated, with or without tides and surge. Or is the flood propagation occurring in a different way? Since flooding in these cells is the main source of the stated improvement in the model when MPAS-O is incorporated, it is important to clarify the flood propagation process in these areas.

Author Response:

We appreciate this comment. To clarify, the ELM–MOSART configuration in Figure 4a does simulate riverine inundation along the mainstem of the Delaware River and some upstream tributaries (as indicated by the red areas aligned with the river network). These inundated areas are generated by excessive precipitation and routed through the river network using MOSART's macroscale inundation scheme, which represents subgrid-scale flooding within individual grid cells. As noted in the Methods section (Section 2.1), this scheme does not explicitly simulate lateral flood propagation between cells, which can lead to some limitations.

However, the lack of inundation near the coastline in Figure 4a is not due to lateral propagation issues, but rather the absence of coastal processes (specifically tide and surge) that are necessary to raise water levels enough for those low-lying coastal cells to become inundated. When MPAS-O is included (Figure 4b), its dynamic two-dimensional wetting and drying scheme enables storm surge and high coastal water

levels to intrude into these shoreline areas, triggering inundation that the ELM–MOSART configuration alone cannot represent. We have clarified this point in the revised manuscript (L310~L316).

"More importantly, although ELM–MOSART simulates extensive riverine inundation along the Delaware River mainstem and tributaries through precipitation-induced runoff (Fig. 4a), it does not capture inundation in low-lying shoreline areas near the coastline. This is because tide and storm surge that elevated local water levels sufficiently to exceed the inundation threshold in coastal cells are not included in this configuration. MOSART's macroscale inundation scheme does not simulate lateral water propagation across grid cells, and coastal inundation requires dynamic oceanic forcing. By integrating MPAS-O (Fig. 4b), which includes two-dimensional wetting and drying, the model captures these near-coastline inundations more accurately."

R2C8:

Line 330-332 and Fig 5: Is the discharge reported as an absolute value? Or is the graph showing the discharge after the coastal water levels have receded and the river begins to flow downstream again? It would be helpful to see the time series of streamflow to understand the temporal effects.

Author Response:

Thanks for the comment. We believe the comment refers to Figure 6, which shows the peak discharge values along the Delaware River mainstem during Hurricane Irene, as noted in the figure caption. The intention of this figure is not to illustrate temporal dynamics, but rather to compare the maximum discharge values among the three model configurations. The elevated peak discharge observed near the river outlet in Experiment 3 is primarily due to the backwater effect induced by high coastal water levels during the storm. While the time series of streamflow and water level are not shown in this manuscript, a full temporal analysis is available in Figure 5 and 7 of Feng et al. (2024). We have clarified this point in the revised text (L353~L355).

"The elevated water levels due to tide and storm surge force an upstream propagation of ocean water into the river channel, resulting in a local increase in peak river discharge and riverine inundation near the outlet, where the highest coastal water levels during Irene lead to elevated maximum discharge values along the lower Delaware River."

R2C9:

Line 442-444: How do the sigma values from Experiment 3 show "the critical need to account for the meteorological uncertainty and is cascading effects through the coupled system"? Please provide more explanation here.

Author Response:

Thanks. We have revised our discussion. Please see our response to R2C2.

R2C10:

Line 473: Exposure implies that there are assets in the flood zone, which I don't think was examined here. "Hazard" is a better word choice.

Author Response:

Thanks. We acknowledge that "risk" is broadly defined, encompassing flood hazard, exposure, and vulnerability (Kron, 2005). Specifically, flood hazard and exposure risks represent the frequency or intensity of flooding events and the extent of human exposure to these events, respectively, as was also discussed in Feng et al. (2023). While we did not explicitly assess the distribution of assets or population, our use of the term "exposure" was intended in a broader sense, referring to the spatial extent of inundation, which implies increased potential for exposure under more severe flood scenarios. To avoid confusion, we have revised the manuscript to replace "flood exposure" with "flood extent" here.

R2C11:

Line 478-479: Why is this the case if Irene was associated with the 75th percentile AMC scenario, as mentioned earlier (Line 252)?

Author Response:

Thanks for the comment. Here we intend to describe that although Hurricane Irene coincided with a relatively wet antecedent soil moisture condition (approximately the 75th percentile), the modeled inundation area still aligns more closely with the best-case scenario, as well as simulations with lower AMCs. This outcome is because reducing soil moisture below Irene's level has a limited effect on peak discharge and inundation, whereas increasing AMC beyond Irene's level (i.e., AMC100) leads to disproportionately larger flood impacts, as shown in Figure 11. This asymmetry is due to the fact that, despite Irene's wet initial conditions, the soil still retained some infiltration capacity at the onset of the storm. In contrast, scenarios with saturated soils (AMC100) overwhelm that capacity, resulting in significantly enhanced runoff and flood extent. We have revised the manuscript to clarify this important point, which we consider a key finding of the study (L503~L509).

"Interestingly, the modeled inundation area for Irene closely aligns with the best-case scenario (Fig. 11b and 11c), despite the fact that Irene occurred under a relatively wet AMC (i.e., 75th percentile AMC). This reflects an asymmetric hydrological response: while drier AMC scenarios show only modest reductions in flood extent, the scenario with saturated soils (AMC100) leads to a disproportionately large increase in peak discharge and inundation. This is likely because, despite the wet soils prior to Irene, there remained sufficient infiltration capacity at the storm's onset. In contrast, further increases in AMC rapidly exceed that capacity, exacerbating surface runoff and flood hazards. This nonlinear amplification highlights the critical role of AMC in modulating compound flood severity."

R2C12:

Line 489: Did the authors confirm that exposure increased? If not, "increasing exposure risks to coastal residents" should be changed to "increasing flood hazards."

Author Response:

We rephrased "increasing exposure risks to coastal residents" to "increasing the extent of flooding, raising potential risks to coastal residents."

R2C13:

Fig 11: In panel (c), what does the Irene scenario (shown in yellow) represent? I thought the purple outline was showing the observed flooding during Irene.

Author Response:

We apologize for the confusion. The purple outline in Figure 11c represents the observed flood extent derived from satellite data (Tellman et al., 2021), consistent to that in Figure 4. The "Irene" scenario shown in yellow corresponds to the best model simulation that represents the Hurricane Irene event at the AMC75 condition, which is also indicated by the dashed lines in panels (a) and (b). We have updated the figure caption to clarify this distinction.

"(c) Spatial map of plausible inundation extents in the DRB, showing the minimum (best-case scenario), Irene (AMC75), and maximum (worst-case scenario) simulated inundation. The purple outline represents the observed flood extent from satellite data, consistent with Figure 4."

R2C14:

Line 496-498: The runtime of the various model configurations was never mentioned, so this statement is unsupported.

Author Response:

We thank the reviewer for pointing this out and have added information on the runtime performance of the simulations to support this statement. Specifically, the global ELM–MOSART simulation required less than 10 minutes using 400 CPUs, while the fully coupled ELM–MOSART–MPAS-O simulation took approximately 5 hours. These runtimes are relatively efficient in ESMs given the global scope and high-resolution coastal refinement in E3SM, and they demonstrate the model's suitability for ensemble-based uncertainty quantification. The relevant sentence has also been revised for clarity.

In the revised Section 2.1, we added the runtime information: "The global ELM–MOSART simulations are computationally efficient, requiring less than 10 minutes using 400 CPUs, while the fully coupled ELM–MOSART–MPAS-O simulations take approximately 5 hours."

In the discussion (L527~L530), the original statement has been rephrased to "Given the global domain and component complexity, the relatively efficient runtime makes the framework suitable for disentangling interconnected drivers of complex physical processes and their cascading effects through ensemble-based analyses."

R2C15:

Line 542-544: "Significantly enhances the accuracy" compared to what baseline? The actual observed flooding was not well predicted by any of the models considered.

Author Response:

Our intention was to convey that the fully coupled E3SM simulation—including the MPAS-O ocean component—demonstrated improved performance in representing compound flood processes compared to the ELM–MOSART configuration alone. However, we agree that the purpose of this framework is not yet to achieve high-accuracy flood prediction relative to fine-resolution regional models. Rather, as discussed in the introduction, this integrated ESM-based framework serves as an intermediate step that enables the identification and analysis of cascading uncertainties across Earth system components. To avoid overstating model performance, we have revised the conclusion text accordingly (L574~L576).

"Our research demonstrates that an integrated atmosphere, land, river and ocean system improves the representation of multivariate flooding processes relative to partially coupled configurations, while enabling the analysis of cascading uncertainties through the multi-component Earth system modeling framework."

Below is the list of newly added references.

Beven, K. and Binley, A.: The future of distributed models: model calibration and uncertainty prediction, Hydrological processes, 6, 279-298, 10.1002/hyp.3360060305, 1992.

Du, X., García-Berthou, E., Wang, Q., Liu, J., Zhang, T., and Li, Z.: Analyzing the importance of top-down and bottom-up controls in food webs of Chinese lakes through structural equation modeling, Aquatic Ecology, 49, 199-210, 10.1007/s10452-015-9518-3, 2015.

Hou, X., Hodges, B. R., Feng, D., and Liu, Q.: Uncertainty quantification and reliability assessment in operational oil spill forecast modeling system, Marine pollution bulletin, 116, 420-433, 10.1016/j.marpolbul.2017.01.038, 2017.

Hu, R., Fang, F., Pain, C., and Navon, I.: Rapid spatio-temporal flood prediction and uncertainty quantification using a deep learning method, Journal of Hydrology, 575, 911-920, 10.1016/j.jhydrol.2019.05.087, 2019.

Naseri, K. and Hummel, M. A.: A Bayesian copula-based nonstationary framework for compound flood risk assessment along US coastlines, Journal of Hydrology, 610, 128005, 10.1016/j.jhydrol.2022.128005, 2022.

Nederhoff, K., van Ormondt, M., Veeramony, J., van Dongeren, A., Antolínez, J. A. Á., Leijnse, T., and Roelvink, D.: Accounting for uncertainties in forecasting tropical-cyclone-induced compound flooding, Geoscientific Model Development, 17, 1789-1811, 10.5194/gmd-17-1789-2024, 2024.

Saleh, F., Ramaswamy, V., Wang, Y., Georgas, N., Blumberg, A., and Pullen, J.: A multi-scale ensemble-based framework for forecasting compound coastal-riverine flooding: The Hackensack-Passaic watershed and Newark Bay, Advances in Water Resources, 110, 371-386, 10.1016/j.advwatres.2017.10.026, 2017.

Santoro, S., Lovreglio, R., Totaro, V., Camarda, D., Iacobellis, V., and Fratino, U.: Community risk perception for flood management: A structural equation modelling approach, International journal of disaster risk reduction, 97, 104012, 10.1016/j.ijdrr.2023.104012, 2023.

Shen, C., Appling, A. P., Gentine, P., Bandai, T., Gupta, H., Tartakovsky, A., Baity-Jesi, M., Fenicia, F., Kifer, D., and Li, L.: Differentiable modelling to unify machine learning and physical models for geosciences, Nature Reviews Earth & Environment, 4, 552-567, 10.1038/s43017-023-00450-9, 2023.

Wang, Z., Leung, M., Mukhopadhyay, S., Sunkara, S. V., Steinschneider, S., Herman, J., Abellera, M., Kucharski, J., Nederhoff, K., and Ruggiero, P.: A hybrid statistical–dynamical framework for compound coastal flooding analysis, Environmental Research Letters, 20, 014005, 10.1088/1748-9326/ad96ce, 2024.